



# Patterns in recent and Holocene pollen influxes across Europe; the Pollen Monitoring Programme Database as a tool for vegetation reconstruction

Vojtěch Abraham[1], Sheila Hicks[2], Helena Svobodová-Svitavská[1, 3], Elissaveta Bozilova[4], Sampson Panajiotidis[5], Mariana Filipova-Marinova[6], Christin Eldegard Jensen[7], Spassimir Tonkov[4], Irena Agnieszka Pidek[8], Joanna Święta-Musznicka[9], Marcelina Zimny[9], Eliso Kvavadze[10], Anna Filbrandt-Czaja[11], Martina Hättestrand[12], Nurgül Karlıoğlu Kılıç[13], Jana Kosenko[14], Maria Nosova[15], Elena Severova[14], Olga Volkova[14], Margrét Hallsdóttir[16], Laimdota Kalniņa[17], Agnieszka M. Noryśkiewicz[18], Bożena Noryśkiewicz[19], Heather Pardoe[20], Areti Christodoulou[21], Tiiu Koff[22], Sonia L. Fontana[23], Teija Alenius[24], Elisabeth Isaksson[25], Heikki Seppä[26], Siim Veski[27], Anna Pędziszewska[9], Martin Weiser[1], and Thomas Giesecke[28]

[1]Department of Botany, Faculty of Science, Charles University; Benátská 2; CZ-128 01; Prague; Czech Republic
[2]P.O. Box 8000, FI-90014 University of Oulu, Finland
[3]Institute of Botany v.v.i.; Czech Academy of Sciences; Zámek 1; CZ-252 43 Průhonice; Czech Republic
[4]Laboratory of Palynology, Department of Botany, Faculty of Biology, Sofia University, 8 Dragan Tsankov blvd., Sofia 1164, Bulgaria
[5]Lab. of Forest Botany, Faculty of Forestry and Natural Environment, Aristotle University of Thessaloniki, P.O. Box 270, 54124 Thessaloniki, Greece
[6]Museum of Natural History Varna, 41 Maria Louisa Blvd. 9000 Varna; Bulgaria
[7]University of Stavanger, Museum of Archaeology, Peder Klows gate 31A, PB 8600 Forus, NO-4036 Stavanger, Norway
[8]Institute of Earth and Environmental Sciences, Maria Curie-Sklodowska University; al. Krasnicka 2d; 20-718 Lublin; Poland
[9]University of Gdańsk, Faculty of Biology, Department of Plant Ecology, Laboratory of Palaeoecology and Archaeobotany, ul. Wita Stwosza 59, 80-308 Gdańsk, Poland
[10]Georgian National Museum, Purtseladze Str.3, Tbilisi 5, Georgia 0105.
[11]Faculty of Biological and Veterinary Sciences, Geobotany and Landscape Planning, Nicolaus Copernicus University in Toruń, 87-100 Toruń, ul. Lwowska 1; Poland
[12]Department of Physical Geography, Stockholm University, SE-106 91 Stockholm, Sweden
[13]Department of Forest Botany, Faculty of Forestry, Istanbul University-Cerrahpaşa; Bahçeköy; TR-34473; Istanbul; Turkey
[14]Depertament od Higher Plants, Moscow State University; Leninskie Gory, 1, 12, Moscow, 119234, Russia
[15]Main Botanical Garden RAS; Botanicheskaya, 4, Moscow, 127276, Russia
[16]Laugarnesvegi 87 íbúð 105, 105 Reykjavík, Iceland
[17]Department of Geography and Earth Sciences, University of Latvia; Jelgavas Street 1, LV-1004; Riga, Latvia
[18]Institute of Archeology, Faculty of History, Nicolaus Copernicus University in Toruń; Szosa Bydgoska 44/48; 87-100 Toruń; Poland
[19]Faculty of Earth Sciences and Spacial Managment, Nicolaus Copernicus University in Toruń; Lwowska 1, 87-100 Toruń; Poland
[20]Department of Natural Sciences, National Museum Wales, Cathays Park, Cardiff, CF10 3NP, U.K.
[21]Department of Forests, Ministry of Agriculture, Rural Development and Environment, P. Box 24136, 1701 Nicosia, Cyprus
[22]Tallinn University, School of Natural Sciences and Health, Institute of Ecology, senior researcher. Uus Sadama 5, 10120 Tallinn, Estonia
[23]Cátedra de Palinilogía, Facultad de Ciencias Naturales y Museo, UNLP, Calle 64 n°3, 1900 La Plata, Argentina
[24]Turku Institute for Advanced Studies (Department of Archaeology), FI-20014 University of Turku





[25]Norwegian Polar Institute, Fram Centre, N-9296 Tromsø, Norway
[26]Department of Geosciences and Geography, University of Helsinki, Gustav Hällströmin katu 2, 00014, Helsinki, Finland
[27]Department of Geology, Tallinn University of Technology, TalTech, Ehitajate tee 5, 19086 Tallinn, Estonia
[28]Palaeoecology, Department of Physical Geography, Faculty of Geosciences, Utrecht University, P.O. Box 80115, 3508 TC Utrecht, The Netherlands.

**Correspondence:** Vojtěch Abraham (vojtech.abraham@gmail.com)

**Abstract.** The collection of modern spatially extensive pollen data are important for the interpretation of fossil pollen diagrams. Such datasets are readily available for percentage data but lacking for pollen accumulation rates (PAR). Filling this gap has been the motivation of the pollen monitoring network, whose contributors monitored pollen deposition in modified Tauber-traps for several years or decades across European latitudes. Here we present this monitoring dataset consisting of 351 trap
locations with a total of 2742 annual samples covering the period from 1981 to 2017. This dataset shows that climate parameters correlating with latitude determine pollen productivity. A signal of regional forest cover can be detected in the data, while local tree cover seems more important. Pollen traps situated beyond 200 km of the distribution of the parent tree are still collecting occasional pollen grains of the tree in question. PAR's of up to 30 grains cm$^{-2}$ y$^{-1}$ in fossil diagram should therefore be interpreted as long distance transport. Comparisons to fossil data from the same areas show comparable values. Comparisons
often demonstrate that similar high values for temperate taxa in fossils sites are found further south or downhill. While modern situations comparable to high PAR values of some taxa (e.g. *Corylus*) may be hard to find, $CO_2$ fertilization and land use may case high modern PAR's that are not documented in the fossil record. The modern data is now publically available in the Neotoma Paleoecology Database and hopefully serves improving interpretations of fossil PAR data.

# 1   Introduction

The interpretation of past vegetation composition from pollen analytical results hinges on the concept of uniformitarianism. Pollen percentages from modern samples of mosses, soil litter and the top sediment of lakes are essential for understanding how the vegetation and other environmental conditions are represented in fossil pollen assemblages (Davis et al., 2013; Jackson
and Williams, 2004; Overpeck et al., 1985). The same understanding is required in the interpretation of pollen accumulation rates (PAR) or the absolute numbers of pollen grains deposited per unit surface area per unit time, only here collecting modern reference values is more difficult. Modern rates of pollen accumulation can be obtained from monitoring pollen deposition using pollen traps (Hicks, 1994) as well as from carefully sampling the top sediments of lakes that are either annually laminated or precisely dated (Matthias and Giesecke, 2014). Monitoring modern absolute deposition of pollen has been conducted since
the development of modern pollen analysis (Giesecke et al., 2010) already allowing Welten (1944) to draw on this information for the interpretation of the first pollen accumulation rate reconstruction from the laminated sediments of Faulenseemoos.



Due to the high interannual variability in pollen production (Andersen, 1980) it is necessary to conduct pollen monitoring over several years to enable comparisons to be made with estimates from sediment cores (Hicks, 1974; Hicks and Hyvärinen, 1999). The network of pollen traps across the latitudinal treeline in Finland (Hicks et al., 2001) demonstrated that these modern analogues can be applied directly to interpret past vegetation changes (Seppä and Hicks, 2006), and this was the motivation for

the establishment of the Pollen Monitoring Programme (PMP, Hicks et al., 1996; 1999; 2001). The program was launched in August 1996 at a meeting in Finland, bringing mainly European researchers together. Members of the network changed over the years and monitoring experiments were discontinued or newly started. Although, pollen monitoring studies were and are carried out on other continents (e.g. Jantz et al., 2013) the PMP had little success in attracting researchers working outside Europe. The standardisation of the monitoring protocol allowed for easy comparisons between the results in different regions

which were discussed at INQUA in 1999 and led to a special volume published in 2001 (Tinsley and Hicks, 2001) collecting results based on several initial time series (van der Knaap et al., 2001; Koff, 2001; Tinsley, 2001; Tonkov et al., 2001) as well as a first comparative study (Hicks et al., 2001). More individual results were published in the following years (e.g. Gerasimidis et al., 2006; Giesecke and Fontana, 2008; Hättestrand et al., 2008; Jensen et al., 2007; Kvavadze, 2001; Pidek, 2007) and comparative studies followed in a second special volume published in 2010 (Giesecke et al., 2010). The data produced by

contributors to the PMP were analysed for different questions, including weather parameters determining the amount of pollen production (van der Knaap et al., 2010) and its correlation to masting years in *Fagus* (Pidek et al., 2010). However, no study has hitherto explored to what extent the PMP data collected provides modern comparisons to fossil situations as was originally intended. Since this dataset extends across the distribution limits of some major European trees, it can also provide information on the quantity of pollen dispersed over long distances, as an analogue for the interpretation of regional or local presence,

which is difficult to achieve based on pollen percentages (Lisitsyna et al., 2011a).

The programme established a database collecting the original data for individual years, as well as general information on the pollen traps installed in the different regions (Fig. 1). The database was developed offline and was thus difficult to access by individual researchers. The paleoecology database Neotoma (Williams et al., 2018) offers a platform to store the PMP data and make it available to researchers worldwide, allowing them to interrogate the data and potentially identify modern

analogues to interpret fossil pollen accumulation rates. The aim of this manuscript is to present an overview of the data in the PMP database and explore its potential to provide analogues for fossil situations. We collected fossil datasets with estimates of pollen accumulation rates from the same regions where the pollen traps were installed to explore the relationship between modern and fossil pollen accumulation rates.

## 2   Methods

### 30  2.1   Study area

Sites in the PMP database were divided into 7 'trap regions' according to the latitude and altitude. These regions were further divided into 'trap areas', by grouping 2-10 trap locations according to their spatial proximity. The arctic/alpine region includes distant trap areas in Spitsbergen and Iceland, northernmost traps in Finland (Utsjoki) and Norway (Lofoten-Vesterålen) and





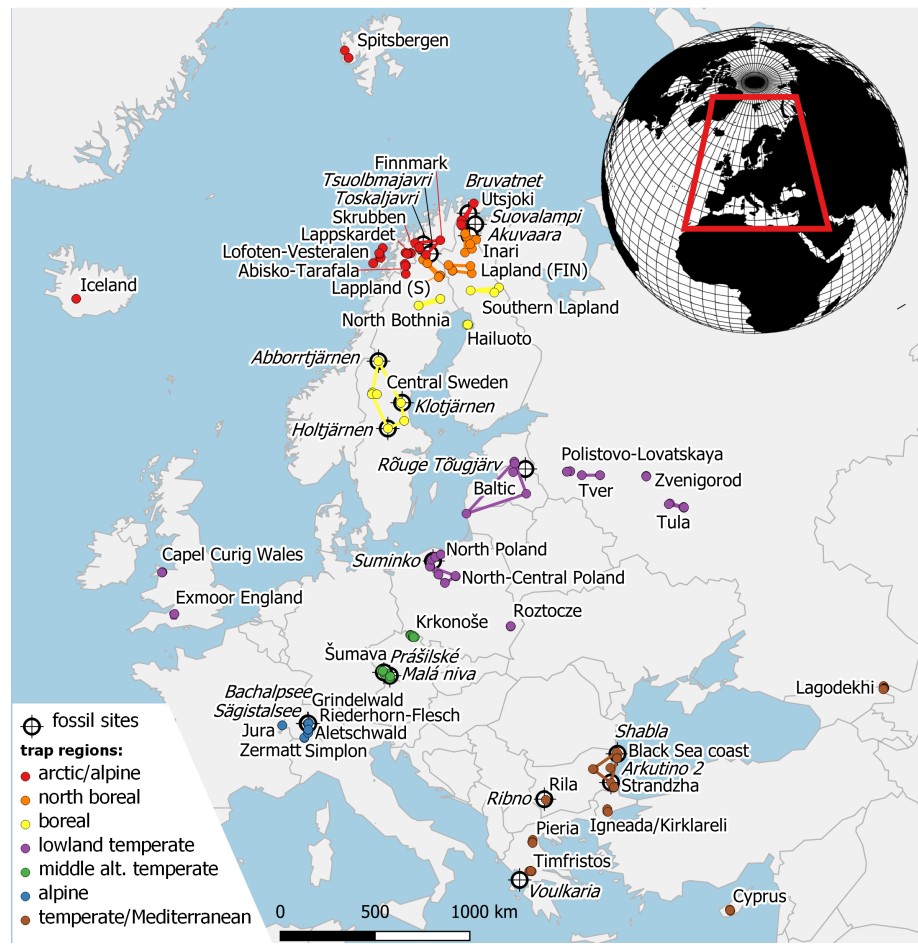

**Figure 1.** Map of the study area. Trap dataset is divided into trap regions (colours) and trap areas (labels). Holocene pollen sites selected for the comparison (target symbol, italic labels – see Tab.1). Colours correspond to Fig. 5.

traps in the Scandinavian mountains above the local tree-line (Finnmark, Abisko-Tarafala, Skrubben and Lappskardet). The landscape around these traps is often treeless or covered by sparse birch shrubland with *Betula nana* and *B. pubescens* in some locations. The north boreal region includes traps in Northern Lapland in Finland, Norway and Sweden with a vegetation dominated by *Betula* and *Pinus sylvestris*. The northern limit of *P. sylvestris* occurs between traps from this region, which

5    are situated at altitudes below 500 m a.s.l. The boreal region includes trap areas situated in Southern Lapland, around the Bay of Bothnia and in Central Sweden. The vegetation is dominated by *Picea abies*, *P. sylvestris* and *Betula* species, with the occurrence of *Alnus incana*. Northernmost populations of *Alnus glutinosa* occur near some sites and the southernmost traps in Central Sweden are situated near northern outpost populations of *Corylus avellana* and *Ulmus glabra*. Traps from the temperate lowland region have the widest longitudinal extent including the British Isles, Poland, the Baltic countries and

10   European Russia. Vegetation at trapping locations below 500 m a.s.l. is characterised by *Quercus* and *Fraxinus excelsior* in





the West and *P. sylvestris*, *P. abies* and *Betula* with an admixture of *Quercus*, *Tilia*, *Ulmus* and *C. avellana* in the East. *Fagus sylvatica* and *Carpinus betulus* occur in Poland, *Abies alba* only in south-eastern Poland. Trapping locations in the temperate region at elevations above 500 m (mid elevation) were separated and include the Krkonoše and Šumava Mountains. Traps in both areas are placed on an elevation gradient from 500 m and 1200 m a.sl. The lower slopes of the mountains are dominated

by *F. sylvatica*, while the traps are situated on a gradient from *P. abies*-dominated forest to the onset of alpine vegetation in Krkonoše. In the Alps and Jura Mountains, traps were placed at even higher elevations, between 1200 m to 3000 m, crossing the altitudinal treeline. Trapping locations in the Southeast represent diverse landscapes and vegetation types including grasslands, evergreen and deciduous forests. Some traps are situated in high mountain regions around treeline situations or within the upper mountain forests including Rila (Bulgaria), Pieria (Greece), Timfristos (Greece), Lagodheki (Georgia) and Cyprus. Traps at

lower elevations are situated near the Black Sea coast, within the low Strandzha Mountains and European Turkey.

## 2.2  Data collection

The pollen traps used in the PMP network generally consist of a bucket or bottle large enough to contain the annual surplus in precipitation on a surface of usually 19.6 cm$^2$ (5 cm diameter opening) or similar. Many traps had a sloping collar inspired by the design of pollen traps by Tauber (1974), although few collars were truly aerodynamic. The collection of the trap content

was generally carried out annually and any special circumstances potentially affecting the annual pollen deposition were noted and stored in the database. For the analyses presented here and data overview we excluded traps where the pollen record is 2 years or less, as averages may be affected by high inter-annual variability. The only exception is the trap situated in Spitsbergen where there is a two-year record. Pollen accumulation from the two-year record shows little variation and, being the only analogue for a truly arctic and treeless environment, provides important information on long distance pollen transport. We

also excluded annual samples with records shorter than 8 months and, in addition, traps or years with spurious values due to particular events or local conditions (Table S1).

Most of the traps in the PMP network were placed in the open vegetation or in forest openings in order to avoid an unrepresentative contribution of individual trees e.g. due to anthers dropping into the traps. Traps were generally installed at ground level mimicking collection conditions relevant for sedimentary archives. In consequence tall herbs or grasses might overgrow

or cover some of the traps potentially leading to higher pollen deposition. Traps not equipped with a mesh occasionally trapped pollen collecting insects, leading to enormous counts of insect pollinated taxa e.g. *Calluna*, *Erythranthe guttata*. The presence of insects in the traps is usually noted for the collection year so that careful evaluation of the information in the database can also inform on herbaceous pollen types (Jensen et al., 2007). Including this information in comprehensive database queries is currently not possible and a manual screening of datasets is required when analysing herbaceous pollen types. This problem

does not seem to occur in tree pollen taxa. The occurrence of phytophagous insects in the traps were not accompanied by unusual peaks in tree pollen taxa, indicating that the insects inadvertently trapped were primarily collecting pollen from the herbaceous vegetation around the traps. Comprehensive database queries restricted to tree pollen, Poaceae and Cyperaceae should therefore not be affected by the occurrence of insects in the trap and mainly represent pollen transport via wind, the rainout of pollen from the atmosphere and the gravity component.



Concentrating the content of the traps was carried out either using filter paper or centrifugation and decanting the supernatant. In many cases the trap content was washed onto a paper filter, which was later digested using acetolysis. Pollen quantity was assessed by adding *Lycopodium* spore tablets (Stockmarr, 1971) to each trap before processing. Pollen concentration was obtained from the ratios between pollen grains counted to *Lycopodium* spike counted and *Lycopodium* spike added. Details

about *Lycopodium* spike data, as well as details of the pollen trap such as the exact size of the opening are stored in the database. The PMP database was created in the PostgreSQL database system. Names of pollen taxa were unified using the accepted variable names from the European Pollen Database (Giesecke et al., 2019).

## 2.3    Investigated taxa and environmental parameters

We selected the common tree and shrub taxa of Europe. Pollen taxa generally refer to all the species within the genus. Pollen

taxa allowing higher taxonomical resolution, which were consistently separated and excluded from the genus in the whole dataset are marked as "excl.". Pollen taxa potentially including pollen grains from another genus are indicated by "incl.": *Abies*, *Alnus* (excl. *A. viridis*), *Betula* (excl. *B. nana*-type), *Carpinus*-type (incl. *C. orientalis/Ostrya*-type), *Corylus*, *Fagus*, *Fraxinus* (incl. *F. ornus*), *Juniperus*-type (incl. *Cupressus*, *Tetraclinis*, *Thuja*), *Picea*, *Pinus* (excl. *P. cembra*-type), *Tilia*, *Quercus* (incl. *Q.robur*-type, *Q. cerris*-type and *Q. ilex*-type). Pollen accumulation rates of trees and shrubs were summed as arboreal

pollen accumulation (hereafter as "tree PAR"). We also included pollen from the plant families Cyperaceae and Poaceae (excluding cereals). For the purpose of the analysis in this paper we refer to sum of tree PAR plus Cyperaceae and Poaceae as "total PAR".

The climate parameters Mean Annual Temperature (MAT) and Annual Precipitation (APrecip) for the trapping locations were obtained from WorldClim 2 (Fick and Hijmans, 2017). For site altitude we used the information supplied by the indi-

vidual investigator. Comparisons between PAR and forest cover were conducted using the data of the Forest Map of Europe (Kempeneers et al., 2012), which has a grid resolution of 1 km$^2$. Forest cover was extracted at a spatial level of all grid cells within a 10 km radius. We used regression analysis to explore whether individual or combinations of these environmental parameters describing the trapping location can explain the variance in average pollen accumulation of the traps. To balance the contribution of high and low pollen producers in the assessment of the total PAR we applied correction factors (Table S2,

Andersen, 1970).

Pollen deposition beyond the distribution area of the parent plant was studied by merging the distribution maps of the relevant species included in the pollen type described above (Caudullo et al., 2018; San-Miguel-Ayanz et al., 2016). These comparisons were not suitable for *Alnus*, *Betula*, Cyperaceae, *Juniperus*, *Pinus* and Poaceae as these taxa are widely distributed in Europe and few traps are located beyond their distribution area. We compared the amount of pollen accumulating from these taxa in

pollen traps at 200 km from their mapped distribution limits. Pollen traps in the UK are situated beyond the natural distribution limits of several of these trees but were excluded from the comparison as the taxa considered may be planted in the area.

For each trap location and each pollen taxon we calculated the distance to the nearest area of distribution using GIS (GRASS Development Team, 2018). Linear regression between this distance and the decadic logarithm of PAR was used to predict the threshold of long-distance transport (hereafter also as "LDT").



## 2.4 Comparison between modern and Holocene PAR

We searched for Holocene pollen records containing estimates of PAR in proximity to the locations of pollen traps. From the

5 available sites we selected at least one and a maximum of three Holocene PAR records per trap region (Table 1). Holocene PAR estimates often show high variation between samples due to changes in the sedimentary environment. To reduce this effect of the here conducted comparison Holocene data were averaged in 500-year bins.

Site and sample compilation resulted in PAR values from 354 Holocene and 271 modern samples. Average trap and fossil PAR values per taxon were submitted to one-dimensional clustering using the R-package Ckmeans.1d.dp (Song and Wang,

10 2011). The classes produced were used to facilitate the comparison between trap and fossil data and to match the modern trap values with analogous situations in the past. The aim of this comparison was to find traps with similarly high values for individual taxa that compared to the highest average fossil PAR. However, averaging did not smooth out all spuriously high values variation.ignoring individual high fossil values as described above (Table S3). Thus, we aimed to find modern analogues for fossil situations represented by several bins (more than 500 years) representing a period of long-term vegetation stability. We linked these periods with high fossil PARs to the closest pollen trap, using a distance matrix between fossil sites and pollen

traps. All statistical analysis and data visualizations were produced in R (R Core Team, 2019).

## 3 Results and interpretations

### 3.1 Spatial pattern of absolute pollen deposition

The PMP database version 02.02.2020 contains data from 351 trap locations with a total of 2742 annual samples covering the period from 1981 to 2017. Trapping sites cover a range of altitudes from 0 to 3000 m a.s.l. with annual precipitation ranging

from 402 to 1549 mm. Mean annual temperature (MAT) for the sites fall between -5.7 to 14.1 °C. The forest cover within a 10 km radius of the trapping sites ranges from 0 to 98%. This range of environmental situations has yielded tree pollen accumulation rates from 5 to 86000 grains cm$^{-2}$ y$^{-1}$, with a median value of 5400 grains cm$^{-2}$ year$^{-1}$ (Fig. 2). An overview of the taxonomic composition of the traps (Fig. S1) shows a dominance of pollen from *Pinus* and *Betula* in the traps from boreal and hemiboreal environments, with *Betula* as the taxon with the highest PARs overall. In most northern traps from open

environments Cyperaceae is the dominant NAP pollen type while Poaceae are dominant in traps from open environments in the south, where they also contribute much higher absolute amounts. The diversity of landscapes and forest types in central and southern Europe is well represented in the pollen composition of traps from this area. Differences in pollen composition and abundance between the high mountain forests of central and southern Europe and the boreal forest in the north are also noticeable. The pollen composition of the fossil sequences selected represents the same regional patterns as the traps (Fig. S2).

Temporal changes are most pronounced in both fossil sequences from the Alps, documenting forest compositional changes as well as the suppression of the tree-line during the Holocene.

Total PAR is generally lower at high latitudes, with the lowest values in the arctic alpine region (trap area Spitsbergen). However, the highest absolute values are not from the southernmost traps but from the lowland temperate region (trap area





**Table 1.** Fossil sequences including type and size of the deposit.

| country | region | site | deposit | latitude | longitude | (m) a.s.l. | area (ha) | reference |
|---|---|---|---|---|---|---|---|---|
| FIN | arctic/alpine | Bruvatnet | lake | 70.17933 | 28.39998 | 119 | 60 | (Hyvärinen, 1975) |
| FIN | arctic/alpine | Toskaljavri | lake | 69.19177 | 21.44841 | 704 | 100 | (Seppä et al., 2002) |
| FIN | arctic/alpine | Tsuolbmajavri | lake | 68.68915 | 22.05235 | 256 | 14 | (Seppä and Weckström, 1999) |
| FIN | north boreal | Suovalampi | lake | 69.58333 | 28.83333 | 104 | 16 | (Hyvärinen, 1975) |
| FIN | north boreal | Akuvaara | lake | 69.125 | 27.68333 | 170 | 4 | (Hyvärinen, 1975) |
| SWE | boreal | Abborrtjärnen | lake | 63.88333 | 14.45 | 387 | 3 | (Giesecke, 2005c) |
| SWE | boreal | Klotjärnen | lake | 61.81667 | 16.53333 | 235 | 1 | (Giesecke and Fontana, 2008) |
| SWE | boreal | Holtjärnen | lake | 60.65 | 14.91667 | 232 | 1 | (Giesecke, 2005a) |
| EST | lowland temp | Rõuge Tõugjärv | lake | 57.73904 | 26.90515 | 114 | 4.2 | (Veski et al., 2012) |
| POL | lowland temp. | Suminko | lake | 53.72556 | 17.77278 | 115 | 0.12 | (Pędziszewska et al., 2015) |
| CZE | middle alt. temp. | Prášilské | lake | 49.07551 | 13.40002 | 1079 | 3.7 | (Carter et al., 2018) |
| CZE | middle alt. temp. | Malá niva | peatbog | 48.90789 | 13.81982 | 754 | 65 | (Svobodová et al., 2002) |
| CHE | mountain temp. | Sägistalsee | lake | 46.68139 | 7.9775 | 1935 | 7.2 | (van der Knaap et al., 2000) |
| CHE | mountain temp. | Bachalpsee | lake | 46.66944 | 8.020833 | 2265 | 8 | (van der Knaap et al., 2000) |
| BGR | temp/medit. | Shabla | lake | 43.58333 | 28.55 | 1 | 1.51 | (Filipova-Marinova, 1985) |
| BGR | temp/medit. | Arkutino 2 | lake | 42.3299 | 27.72363 | 0 | 40 | (Bozilova and Beug, 1992) |
| BGR | temp/medit. | Ribno | lake | 42.20682 | 23.32346 | 2184 | 3.5 | (Tonkov et al., 2002) |
| GRC | temp/medit. | Voulkaria | lake | 38.866667 | 20.833333 | 0 | 10000 | (Jahns, 2004) |

Tver; Fig. 2). Nevertheless, latitude alone explains about 11% of the variance in log transformed total PAR, while MAT and
forest cover within 10 km explain 21% and 72% respectively. In combination, these three variables explain 76% and with
addition of elevation 73% of the total variance in log transformed total PAR. Adjusting the PAR from individual taxa by
Andersen factors reduces the bias of differential pollen production between different plants and thus increases the amount of
variance explained by the regression model with all 4 explanatory variables to 82% (Table S4). This adjustment reduced the
individual explanatory power of forest cover within 10 km, due to the inclusion of grasses in the total PAR. On the other hand,
latitude alone explains 38% of the Andersen adjusted log transformed total PAR (Fig. 3a).

The regression models consider the full range of the data, while due to local factors there is often a spread of average trap
values for different traps in the same region. Traps with maximum values per region do not follow a latitudinal pattern, the
distribution of the minimum average trap values are more informative (Fig. 3a). These lower values closely follow a latitudinal
trend. The average PAR south of 62° latitude and below the altitudinal treeline or close to forests are generally higher than
1000 grains cm$^{-2}$ y$^{-1}$. An area with low PAR in the south is the coastal grassland in northern Bulgaria. The generally low PAR
in this area can be explained by the sparse vegetation cover on thin rendzina soils formed on limestone rock. Adjusting the
PAR values by Andersen factors increases the values for this region so that they fit the general latitudinal trend. (Fig. 3a). Traps



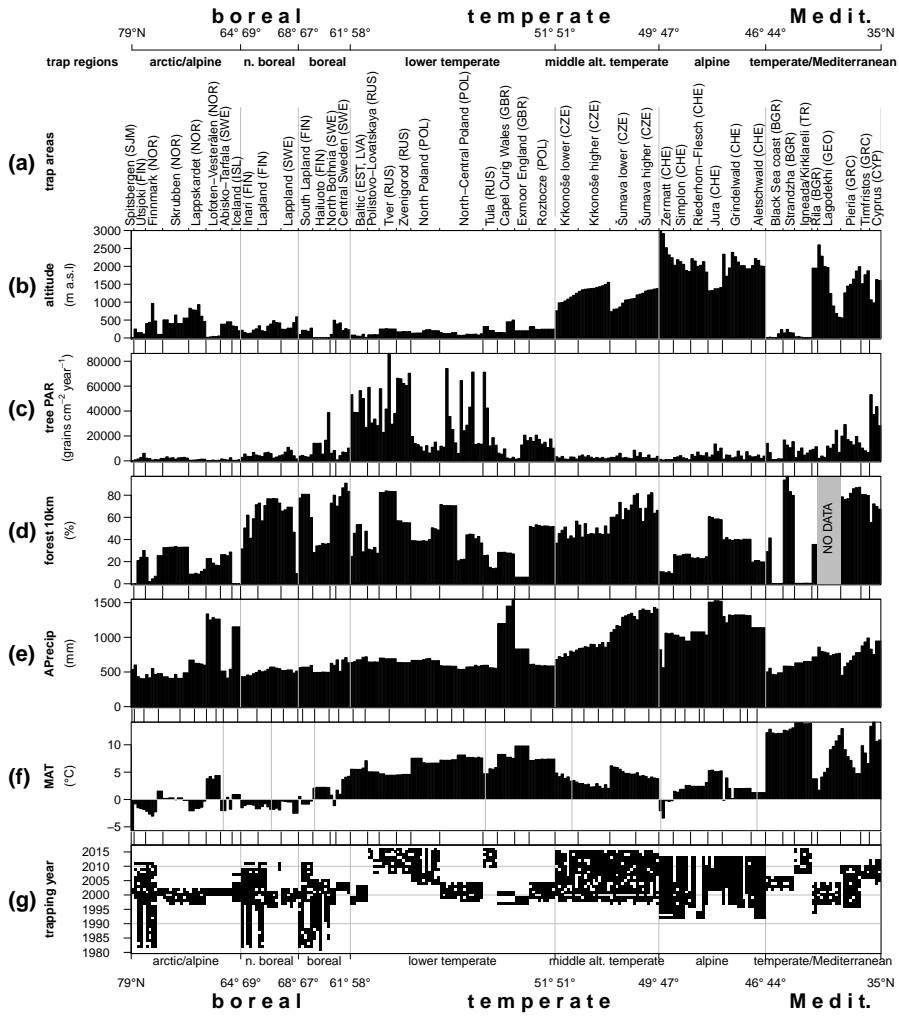

**Figure 2.** Environmental setting of the trap dataset. (a) trap areas ordered from north (left) to south (right), (b) altitude, (c) mean annual tree PAR, (d) forest cover within 10 km radius, (e) annual precipitation, (f) temperature: MAT - Mean Annual Temperature, (g) temporal coverage of the PMP database.

5    with minimum average PAR values per region also correspond well to the forest cover within 10 km (Fig. 3b). Taking the traps with minimum PAR within 3% wide bins of the forest cover data provides a regression model predicting a tree PAR of 3200 grains cm$^{-2}$ year$^{-1}$ at 80% forest cover within 10 km of the trap.

The comparison of PAR with the distribution limit of different tree taxa shows that PAR generally decline with distance (Fig. 4). A gradual decline is best documented for *Quercus* where average values at the distribution limit scatter around 100 grains cm$^{-2}$ y$^{-1}$. This analysis also documents the long-distance transport of many tree pollen, including the heavy pollen of *Picea*. Where pollen numbers are very low in this comparison there is a degree of uncertainty since the likelihood of encountering a

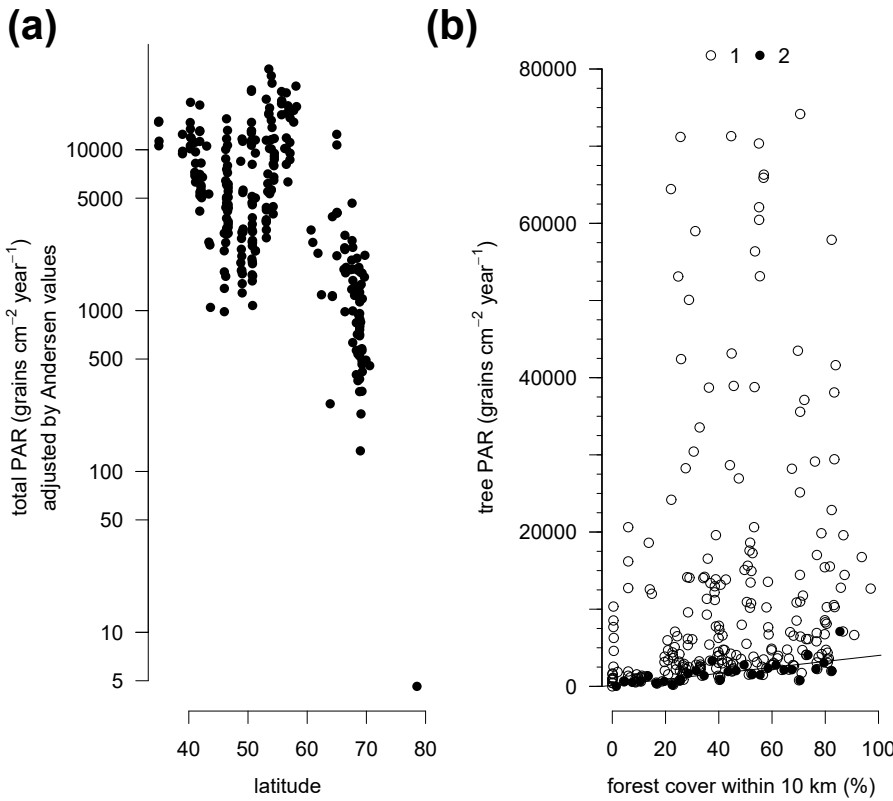

**Figure 3.** Latitudinal gradient in pollen accumulation rates of major tree taxa and Poaceae and Cyperaceae (total PAR) adjusted by Andersen values (see Tab S2) of pollen representation (a). Relationship between forest cover within 10 km radius and tree PAR (b). All trap sites (1) and minimum tree PAR per every 3 % of forest cover (2).

grain depends on chance and is related to the number of pollen grains counted, which we did not consider in this analysis. For better comparison of the absolute values between taxa we used regression analysis to estimate the amount of pollen at 200 km from the distribution limit (Fig. 4b). This comparison indicates that less than 30 grains cm$^{-2}$ y$^{-1}$ of *Carpinus, Corylus, Fagus, Fraxinus, Quercus* and *Tilia* are deposited beyond 200 km of the distribution of the parent trees. Only *Picea* shows less than 1 grain cm$^{-2}$ y$^{-1}$ at 200 km of the distribution range.

## 3.2 Recent and fossil PAR values at continental and regional level

Comparison of modern and fossil PAR values for the dataset presented here produces good agreement with similar peaks in total PAR between 2000 and 10000 grains cm$^{-2}$ y$^{-1}$ (Fig. 5). Maximum PARs in the pollen traps are always higher (often ten times higher) compared to fossil situations, with the exception of *Corylus*. The frequency distribution of PARs is log normal for *Alnus*, *Tilia* and *Fraxinus*. A bimodal distribution of values occurs for some taxa, which is particularly clear for the fossil values of *Abies*, *Picea* and Poaceae. In the modern samples such a bimodality can be recognized for *Pinus* with a trough at



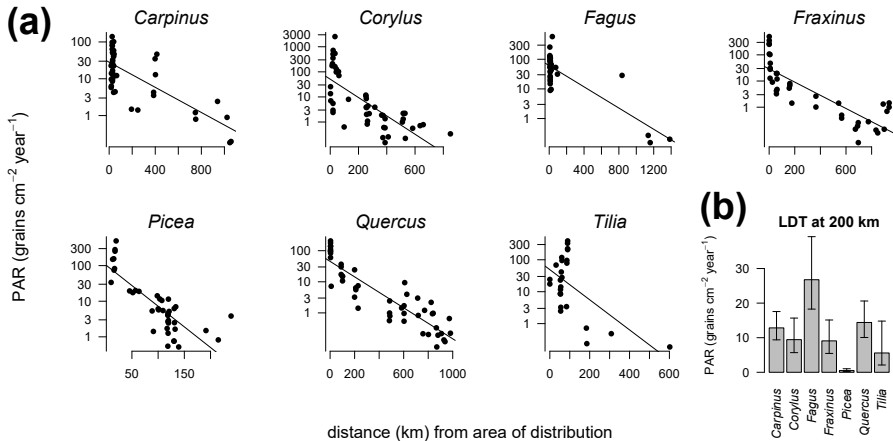

**Figure 4.** Relationship between the distance from the trap site to the nearest area of species distribution and PAR for selected trees (a). PARs of the long-distance transport (b) calculated from linear regression at 200 km (Fig. 4a). Traps within the area of species distribution were excluded.

5    around 1000 grains cm$^{-2}$ y$^{-1}$. The frequency distribution of modern and fossil PARs corresponds best for *Corylus*, with most values falling between 100 - 300 grains cm$^{-2}$ y$^{-1}$ and recent and fossil maxima at around 3000 grains cm$^{-2}$ y$^{-1}$. The greatest difference in the distribution of modern versus fossil PARs occurs for *Juniperus*, where maximum values are around a hundred times larger in the traps. Minimal PARs are about ten times higher in the traps for Poaceae and Cyperaceae in particular and the right side of the distribution is shifted upwards. For *Fagus, Quercus* and *Carpinus* the fossil PARs show a local maximum in

10   the frequency of low values, which does not occur in the traps. These frequent low values range below the threshold indicating long distance transported pollen.

There is a poor correspondence between modern and Holocene PARs for the individual trapping regions. Using the 15 selected taxa in seven trap regions and the occurrence of the pollen types we obtained 92 pairs; of these 31 pairs are similar based on a t-test and a p-value > 0.05 (Fig. 6). In this regional comparison *Betula* shows the best agreement between modern and

15   fossil values. Values are similar in four regions across the gradient, with highest values of 5400 grains cm$^{-2}$ y$^{-1}$ in the lowland temperate region and the lowest values of 34 grains cm$^{-2}$ y$^{-1}$ in the temperate/Mediterranean region, where the parent trees are generally absent. Modern and fossil PARs generally correspond well for the lowland temperate region where, in addition to *Betula*, *Alnus*, *Carpinus*, Cyperaceae, *Fraxinus*, *Picea*, *Pinus* and *Quercus* also have similar values. Although *Corylus* has a good overall agreement, the regions with similar modern to fossil data are shifted, with Holocene values in the boreal region corresponding to modern PARs in the lowland temperate region.

**3.3    Recent and fossil PAR values at the levels of sites, traps and trap areas**

The one-dimensional cluster analysis distinguished between 5 and 9 classes of PAR values per taxon. Comparing the maximum averages of fossil PARs to modern trap data on a site by site basis shows that it is possible to find modern comparisons for all

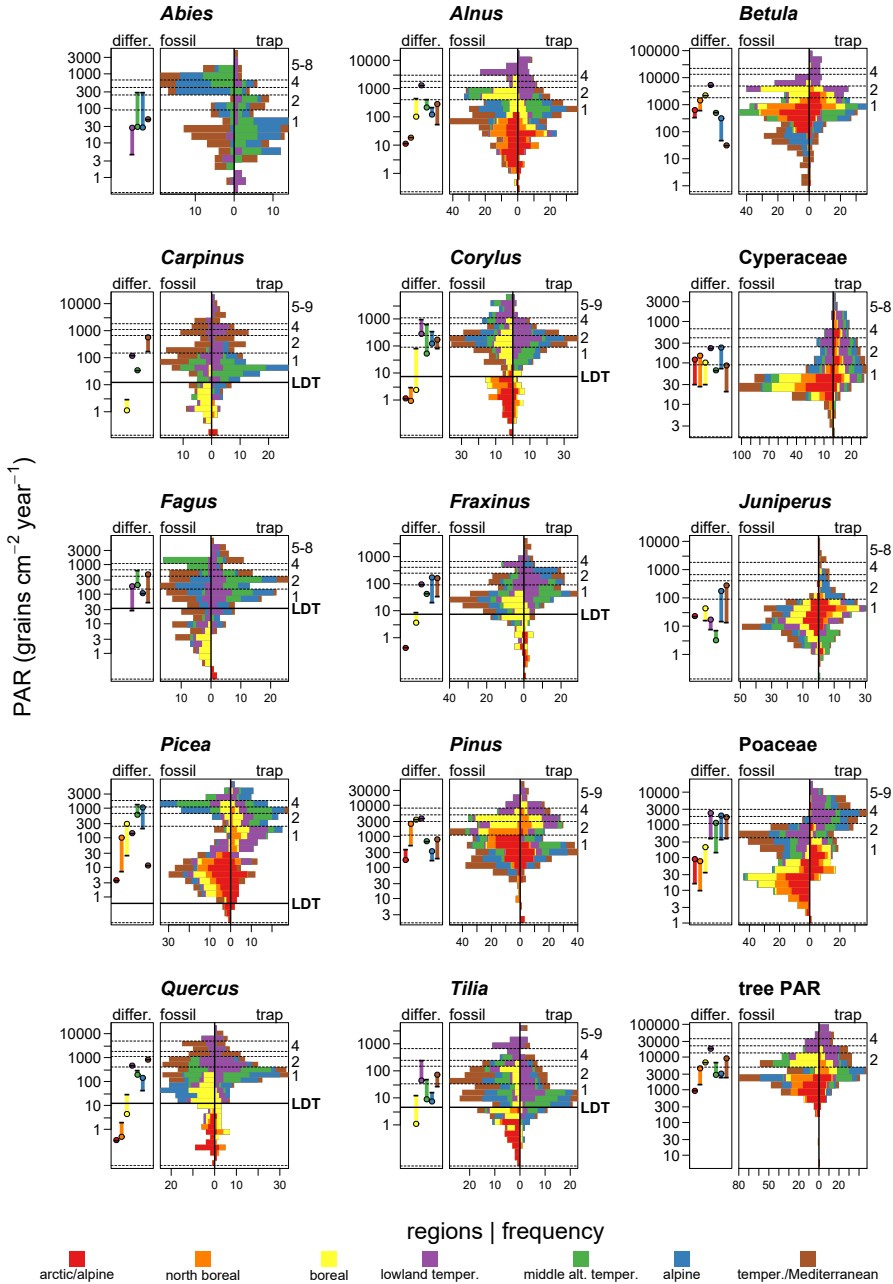

**Figure 5.** Difference (differ.) between the mean fossil (-) and the mean trap (o) pollen influx per trap region is shown by length of the vertical segment. Paired histograms of mean annual pollen influx from fossil record (on the left) and from traps (on the right). Colours denote different trap regions and correspond to Fig. 1. Note logged y-axis. Horizontal lines and numbering on the secondary y-axis denote classes of PAR, for more detail see Fig. 6b) and d). LDT is threshold for long distance transport.



fossil situations. Fossil PARs from sites in the arctic and boreal region often match with modern values in pollen traps situated south of the sites. The frequently high modern values in pollen traps from northern and central Poland in particular provide frequently good matches for fossil situations. The spatiotemporal pattern for *Juniperus* stands out, with highest fossil values in the Early Holocene and core top samples having analogues at the latitudinal and altitudinal treelines. We demonstrate the linkage of the highest PAR clusters per each fossil site with individual trap records on example of main taxa (*Abies*, *Betula*,

*Corylus*, *Fagus*, *Picea*, *Pinus*, *Quercus*, *Tilia*; Fig. 6). Detailed description of the rest of taxa see (*Alnus*, *Carpinus*, Cyperaceae, *Fraxinus*, *Juniperus*, Poaceae, arboreal pollen; Fig. S3).

### 3.4   Taxa specific linkage of the highest average PAR at fossil sits with individual trap values

#### 3.4.1  *Abies*

Modern PAR of 490-3900 grains cm$^{-2}$ y$^{-1}$ are observed in Roztocze, Jura, Rila and Timfristos. These values are produced

by the different species: *Abies alba* in Roztocze and the Jura mountains and *A. cephalonica* in the Rila mountains and on Timfristos. Generally high fossil values occur in the two Alpine lakes Sägistalsee and Bachalpsee around 7000 years, in Ribno in the Rila mountains around 5000 years and in the two sites in Šumava between 4000 and 1000 years. In all these regions with fossil evidence of high *Abies* populations the modern PAR values in pollen traps are comparably low documenting that the populations have much declined. The *Abies alba* populations in Roztocze provide modern analogues for how dense *Abies*

forests may have been in the Alps and Šumava although the forests in Roztocze occur at much lower elevations. Pollen traps on Timfristos mountain provide analogues of the density of Middle Holocene *Abies* forests in the Rila mountains. Pollen traps located far from the distribution limits of *Abies* in Wales, northern Poland, Georgia and Cyprus registered occasional *Abies* pollen grains with PARs of up to 80 grains cm$^{-2}$ y$^{-1}$.

#### 3.4.2  *Betula*

Ignoring traps from the Caucasus and Turkey, trap sites cover the distribution of *Betula pendula* and *B. pubescens*. The highest modern values between 11900-73900 grains cm$^{-2}$ y$^{-1}$ are found in Estonia and Russia as well as in one pollen trap from Hailuoto. As discussed in the main manuscript these modern PARs exceed values from fossil examples. The highest fossil PAR of around 10000 grains cm$^{-2}$ y$^{-1}$ in this comparison come from northern Poland and Estonia and are thus consistent with the area of high modern values. Whether high abundance of *Betula* is a characteristic of the eastern European forests or a result of

frequent disturbance at the forest ecotone or due to recent land-use change is difficult to evaluate based on the available data. Modern and fossil values agree for the sites in central Sweden at PARs between 1900-5600 grains cm$^{-2}$ y$^{-1}$.

#### 3.4.3  *Corylus*

Modern PARs stay below 2800 grains cm$^{-2}$ y$^{-1}$ except for two traps in north-central Poland. Values above 610 grains cm$^{-2}$ y$^{-1}$ are also found in pollen traps from the Baltic, Wales, Turkey and Georgia. The pollen type mainly comes from *Corylus avellana*, while *C. maxima* occurs in Greece, *C. colchica* in Georgia and *C. colurna* in Turkey and in plantations or as ornamental trees





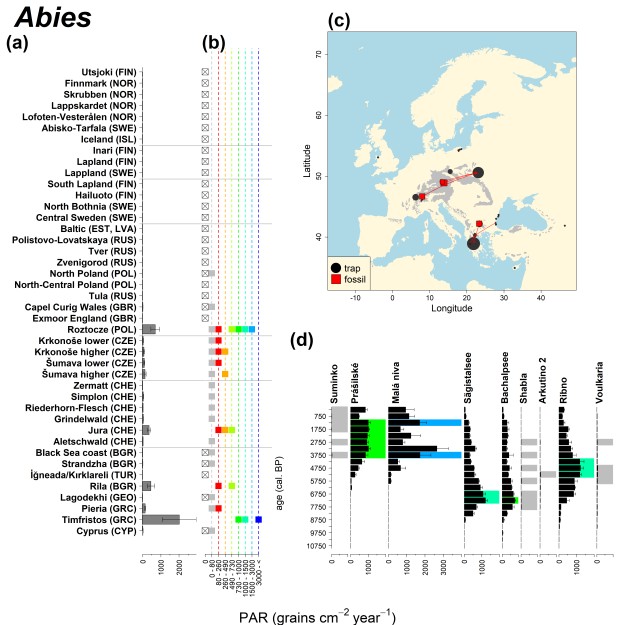

**Figure 6.** Mean modern PARs averaged for each trap area (a). b) Range of mean individual trap values classified by one-dimensional clustering. Crossed squares indicate that pollen of the taxon was not found in any trap from the area. c) Map of Europe with the distribution of the species (Caudullo et al., 2018, San-Miguel-Ayanz et al. 2016) falling within the pollen taxa, size of symbols shows classes of PAR in recent and the highest PAR per each fossil record. Arrows show the closest trap with the same class of PAR. d) Fossil PAR values with the highest PAR class per each record highlighted by the corresponding colour for the class (see b) Note the scale of the x-axis corresponds to the x-axis scale of graph a).

in the rest of the Europe. As discussed in the main text, the selection of fossil sites did not include studies from western and lowland central Europe where values of 10000 grains cm$^{-2}$ y$^{-1}$ are common for the Early Holocene. The highest fossil PARs from the chosen examples were estimated for the Early Holocene from Prášilské situated at 1000 m a.s.l. Thus the high modern values in north-central Poland provide analogues for several fossil situations. Occasional grains and small PARs of *Corylus* pollen are common in traps from boreal regions as well as two traps from the arctic region.

### 3.4.4 *Fagus*

PARs of more than 3300 grains cm$^{-2}$ y$^{-1}$ are found in pollen traps in Poland and Georgia and values above 2100 occur in traps from the Strandza, Šumava and Jura mountains, in southern Bulgaria, Czechia and Switzerland respectively. Two species of *Fagus* contribute to the pollen type with *Fagus sylvatica* as the dominant tree across much of Europe and *F. orientalis* occurring in southern Bulgaria, Turkey and Georgia. Fossil PAR increase around 8000 cal. BP at the Bulgarian Black Sea coast, around 7000 in the Šumava Mountains and 1500 years ago in northern Poland. High values in Šumava and northern Poland match the values in adjacent traps. Hardly any *Fagus* pollen occurs in pollen traps outside its modern distribution except sporadic



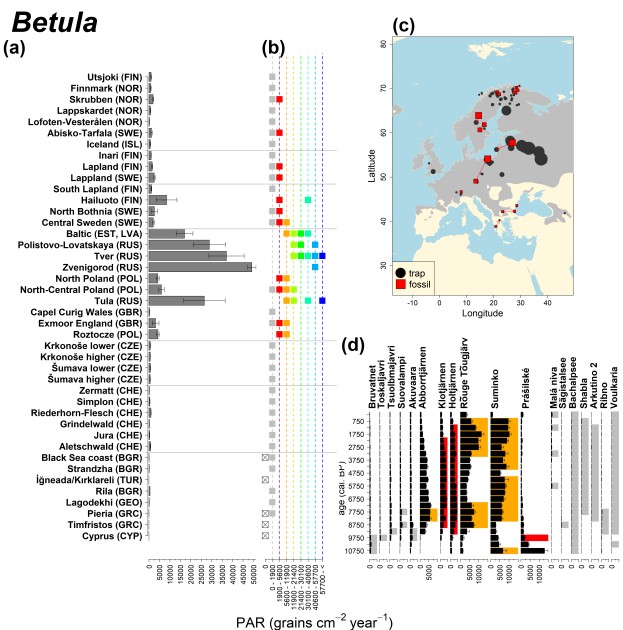

**Figure 7.** See caption Fig. 6

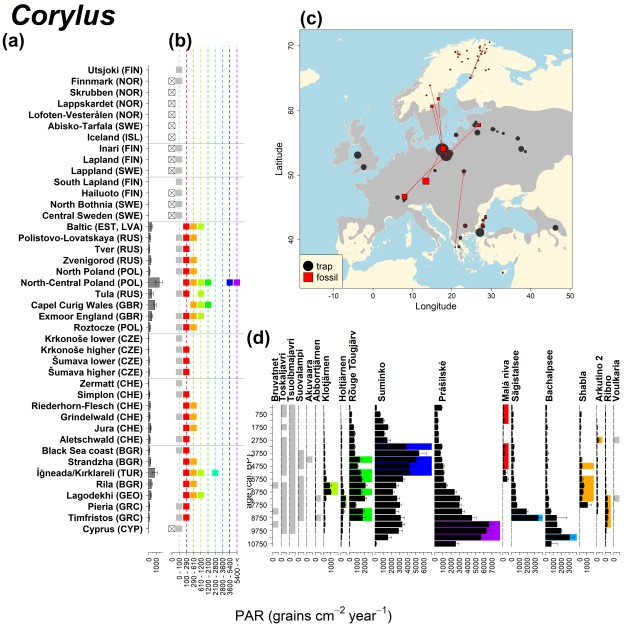

**Figure 8.** See caption Fig. 6





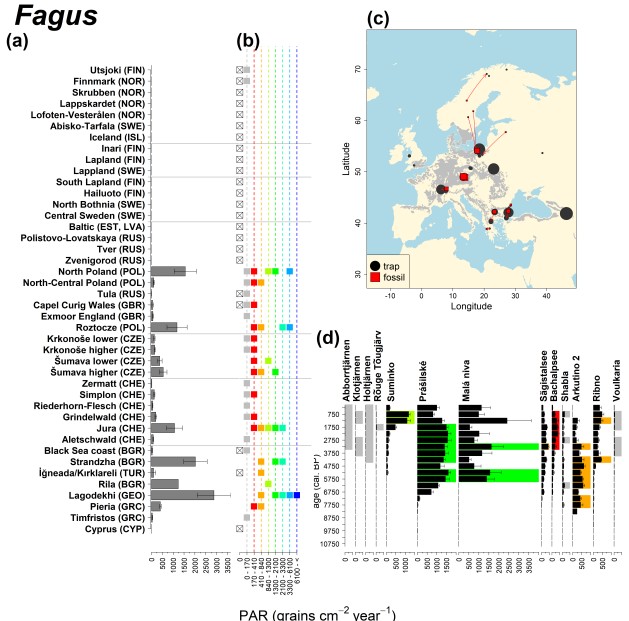

**Figure 9.** See caption Fig. 6

appearances in Tula and the two northernmost trap areas. However, *Fagus* pollen occurs regularly at fossil sites that were assumed to have never been within the distribution of the tree, such as the sites in central Sweden, where *Fagus* grains occurred regularly over the last 3000 years.

### 3.4.5 *Picea*

Modern *Picea* PARs above 2800 grains cm$^{-2}$ y$^{-1}$ are present in traps from Central Sweden, the Baltic region, Russia, Šumava mountains and the Alps. While some of the modern *Picea* pollen may be released by planted non-native *Picea sitchensis* and *P. pungens*, most of the pollen comes from Picea abies, which is also planted in many European regions outside its natural distribution. The highest fossil values in the selection of sites come from the Šumava mountains and are comparable to high trap values from the same region although at lower elevations. Noteworthy are also the generally low fossil Picea PARs for sites in Central Sweden and Tver region, where the tree is dominating or co-dominating the forest for the last 2000 years. Most pollen traps from beyond the distribution area of Picea collect individual Picea pollen.

### 3.4.6 *Pinus*

Highest modern PARs exceeding 43600 grains cm$^{-2}$ y$^{-1}$ are observed in traps on Cyprus, while the values in traps from the northern boreal forest often stay below 5400 grains cm$^{-2}$ y$^{-1}$. *Pinus* PAR values increase from Finnmark (2000 grains cm$^{-2}$ y$^{-1}$) and central Sweden (5000 grains cm$^{-2}$ y$^{-1}$) to the Baltic and north-central Poland with 35000 grains cm$^{-2}$ y$^{-1}$. In northern Europe nearly all *Pinus* pollen comes from *P. sylvestris*, while southern European trapping sites have a higher diversity of trees within





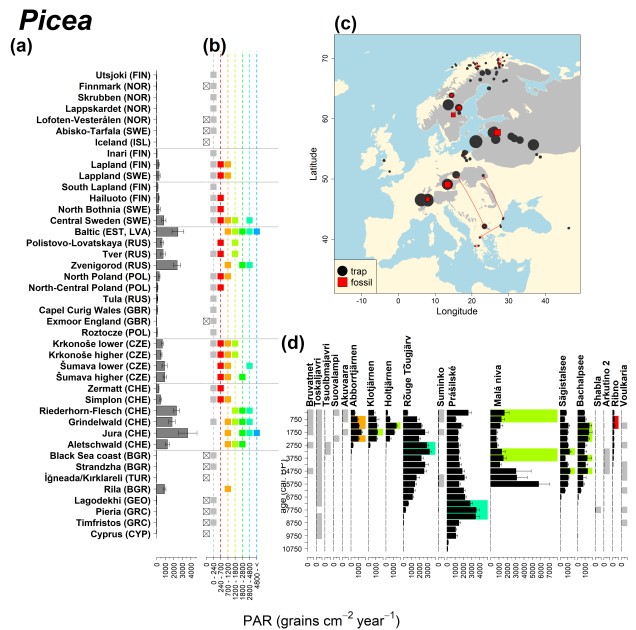

**Figure 10.** See caption Fig. 6

the subgenus Diploxylon including *P. mugo*, *P. nigra*, *P. brutia* and *P. halapensis*. Pollen of both Haploxylon pines *P. cembra* in Alps and *P. peuce* in Rila was separated. High fossil *Pinus* PAR values are estimated for Suminko in northern Poland, matching modern trap values from the same region. In the Šumava mountains *Pinus* was possibly the dominant forest tree during the Early Holocene and declined thereafter, so that modern values from north-central Poland provide the nearest analogue to the Early Holocene situation. The lowest *Pinus* PARs (<125 grains cm$^{-2}$ y$^{-1}$) are found in taps from Iceland (2-5 grains cm$^{-2}$ y$^{-1}$), Lagodheki and most of the traps from Lofoten-Vesterålen (38-125 grains cm$^{-2}$ y$^{-1}$) and single traps from Lappskardet, Exmoor England and Zermatt.

### 3.4.7 *Quercus*

Modern PARs of *Quercus* within the area of distribution of a parent tree in the genus range between 620-15000 grains cm$^{-2}$ y$^{-1}$. Highest modern values (6000-15000 grains cm$^{-2}$ y$^{-1}$) are found in the traps from the UK, Poland and the southern Balkan. The first two areas host only species belonging to the *Q. robur*-type, whereas the two latter also include species from *Q. cerris*-type and *Q. ilex*-type. The highest fossil values from Suminko seem too high for the region compared to other fossil samples, nevertheless modern values in pollen traps from the region provide comparable high PARs. Also, PARs in recent sediments from some lakes in north-eastern Germany show similar values (Matthias and Giesecke, 2014). PARs to about 1300 grains cm$^{-2}$ y$^{-1}$ are estimated for Rõuge Tõugjärv near the distribution limit of *Quercus robur* where the tree only became abundant after 7000 years ago. *Quercus robur* reached its maximum abundance even later in central Sweden where the PAR at Holtjärnen around 3200 years ago suggest its presence. The pollen seems to disperse well and is found in small amounts in most pollen





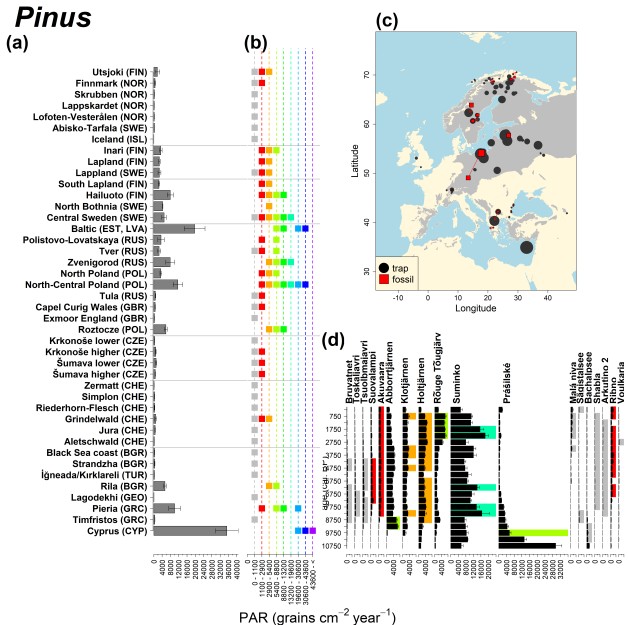

**Figure 11.** See caption Fig. 6

traps beyond its distribution area and fossils samples far to the north of the distribution have also collected *Quercus* pollen during the Holocene, which may be partly due to the abundance of trees of this genus in Europe.

### 3.4.8 *Tilia*

The highest modern PAR 1500-4700 grains cm$^{-2}$ y$^{-1}$ was measured in Poland, European Russia (Tula) and Lagodheki in Georgia, however, this high range has no comparison in the fossil record. Lower values 120-1500 grains cm$^{-2}$ y$^{-1}$ were found in traps from European Turkey and the Alps. Highest fossil PAR 270-1500 was measured in middle altitude and lowland temperate zone during the Middle Holocene. The closest trap analogues to them are in North Poland. Sites on the current edge of *Tilia* distribution and in the mountains (Central Sweden, Alps and Rila) show highest PAR range 120-270 grains cm$^{-2}$ y$^{-1}$, also during the Middle Holocene. Trap records corresponding to the lowest class and above the LTD threshold (5-40 grains cm$^{-2}$ y$^{-1}$) appear in sites within distribution limit of *Tilia* (Baltic, European Russia, Poland), on its edge (Central Sweden, Wales, Black Sea Coast, Greece) or in the mountains (Šumava, Alps).



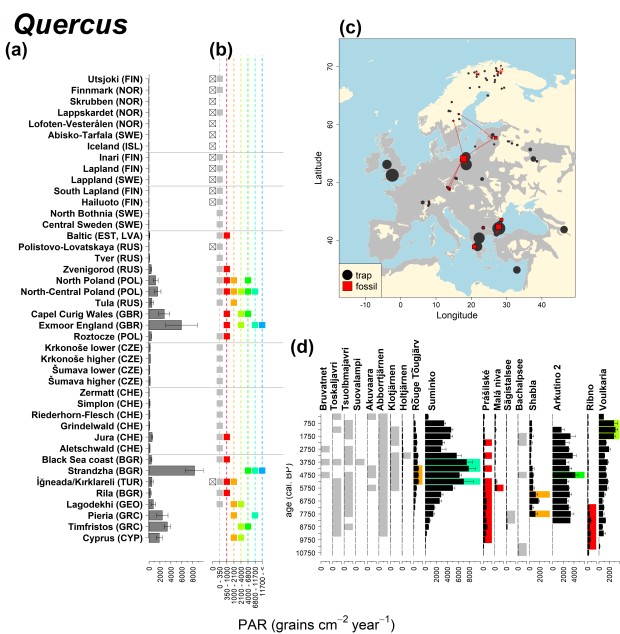

**Figure 12.** See caption Fig. 6

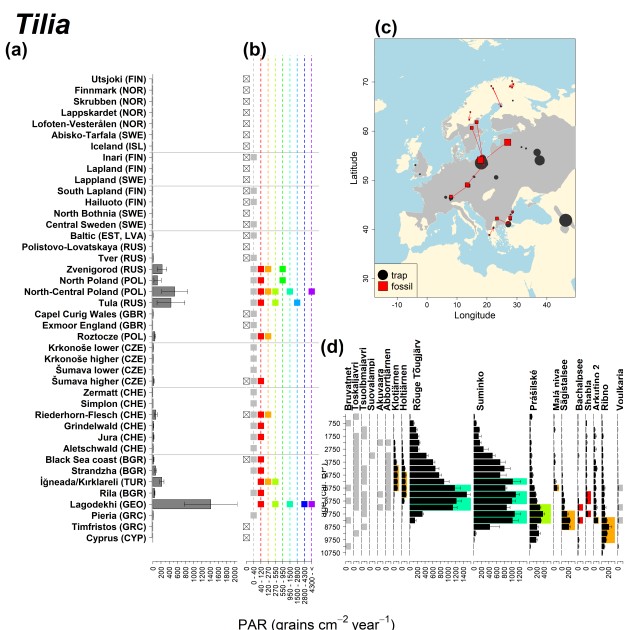

**Figure 13.** See caption Fig. 6





## 4 Discussion

### 4.1 Analogues for vegetation reconstruction

This overview of European pollen trap data collected by the PMP network demonstrates that modern PARs provide comparable values to fossil records and can thus help interpreting the fossil signal. The dataset extends across the European latitudinal and altitudinal range and even if it lacks representation of western European vegetation types, it documents some general patterns. The latitudinal gradient in PAR is clearly visible in this dataset. Although data on plant biomass and primary productivity are not available for all trapping locations the regression analysis indicates that mean annual temperature has a strong influence on the quantity of pollen deposition. The July temperature of the previous year determines the amount of pollen production in *Pinus* near the tree-line (Autio and Hicks, 2004; McCarroll et al., 2003). Evidence from other European regions (van der Knaap et al., 2010; Nielsen et al., 2010) suggests that growing season warmth and other climate variables also explain the interannual variability of pollen deposition. On the regional scale PAR corresponds to plant biomass of the parent tree (Matthias et al., 2015; Seppä et al., 2009). However, differences in plant biomass cannot explain the latitudinal gradient in PAR described here, which may, at least in part, result from the latitudinal gradient in primary productivity of trees (Gillman et al., 2015) as previously suggested (Matthias et al., 2015). An increase in primary productivity and pollen production has been shown in a carbon dioxide fertilization experiment (Wayne et al., 2002), which supports the interpretation that PAR of the same species may vary due to environmental parameters determining its productivity.

Modern PARs from traps near the latitudinal limit of *Pinus* and *Betula* have been used previously to reconstruct past changes in the northern distribution limits of these trees (Seppä and Hicks, 2006). Here we evaluated larger distances and therefore had to ignore *Pinus* and *Betula*, while suggesting some general thresholds for other dominant European trees. The upper value of around 30 grains cm$^{-2}$ y$^{-1}$ for *Picea* agrees well with the fossil PAR value for the tree of 50 *Picea* grains cm$^{-2}$ y$^{-1}$ found in a sample at Klotjärnen just after the occurrence of the first *Picea* bud scale (Giesecke, 2005b). However, these modern thresholds estimated here are likely to depend on the abundance of the parent tree in the larger region rather than properties of the pollen types. A larger threshold would be expected for *Corylus* compared to *Fagus*, based on the fall speed of pollen. However, *Corylus* is not very abundant near its northern limit in these more continental areas, while *Fagus* often dominates forests near its presumed limits. Similarly, the occurrences of taxa outside the mapped natural distribution can bias this estimate.

Linking the fossil to modern PAR values facilitates interpretation of the fossil record of individual sites. Unfortunately, the details cannot be discussed here. However, the central Swedish sites Holtjärnen and Klotjärnen provide excellent examples. These sites are situated north of the modern distribution of *Tilia*, *Corylus*, *Quercus* and near the limit of *Alnus glutinosa*. The fossil PAR values are higher for these taxa than those found in pollen traps at or near these lakes (Giesecke, 2005a; Giesecke and Fontana, 2008). Modern reference values for the PARs of these taxa can be found in northern Poland and Estonia. Moreover, this analogue matching indicates that 3000 years ago the PAR values for *Quercus* at Holtjärnen were sufficiently high to indicate the occurrence of small populations near the lake.

While the dataset of modern PARs presented here provides analogues for the selection of fossil sites, reliable fossil PAR records were not available from the trapping locations in the UK. Moreover, no sites with fossil PAR from western Europe or





from low elevations in the Alps are included in this comparison. Consequently, the large quantities of *Corylus* pollen deposited in many west European sites during the early Holocene was not considered. There are limited modern analogues for the highest early Holocene values of around 7000 *Corylus* grains cm$^{-2}$ y$^{-1}$ at Prášilské. Average Early Holocene *Corylus* PAR at Soppensee in northern Switzerland are 12000 grains cm$^{-2}$ y$^{-1}$ (Lotter, 1999) and at Meerfelder Maar (Kubitz, 2000) in western Germany

18000 grains cm$^{-2}$ y$^{-1}$. Judging from pollen percentages even higher Early Holocene values should be found in more oceanic situations and the *Corylus* PAR at Hockham Mere in eastern England may be as high as 40000 grains cm$^{-2}$ y$^{-1}$ for the early Holocene (Bennett, 1983). Modern values in pollen traps from Wales at around 2000 grains cm$^{-2}$ y$^{-1}$ are far below these early Holocene figures and it is likely that modern analogues of sites with high *Corylus* PARs no longer exist in Europe.

Conversely, the high modern PAR values for *Pinus* and *Betula* from Poland and Latvia are not found in the fossil examples.

*Pinus* PAR values around 30000 grains cm$^{-2}$ y$^{-1}$ were also obtained from $^{210}$Pb dated modern lake sediment samples in north eastern Brandenburg (Matthias and Giesecke, 2014). This study evaluated the PARs for the years 1993 and 2009. The increase in *Pinus* PAR values between the first and the second sampling period corresponded with an increase in the amount of standing pine volume in the region. Forestry practices aimed at increasing yield could account for the high *Pinus* values. *Pinus* was extensively planted after the 1950s, even on soils where trees with a lower pollen production would have grown

naturally. The fertilization due to increased nitrogen deposition, as well as increased atmospheric carbon dioxide, increase the pollen production not only of *Pinus*. A carbon dioxide enrichment experiment of 19-year old *Pinus taeda* resulted in a twofold probability of reproductive maturity after 3 years (LaDeau and Clark, 2001). The continued experiment also showed that carbon dioxide fertilization increased the number of pollen cones and therefore pollen grains produced per tree (LaDeau and Clark, 2006).

In the case of *Pinus*, the modern dataset includes trap data from Cyprus, where *Pinus brutia* dominates at 1600 m a.s.l., resulting in even higher *Pinus* PAR values compared to those found in the Polish and Baltic regions. The highest *Betula* values come mainly from Russia, where values frequently exceed 30000 grains cm$^{-2}$ y$^{-1}$. We previously discussed such high fossil PAR values for *Corylus*, which is assumed to produce a similar amount of pollen. However, fossil *Betula* PARs in the examples considered here are consistently below 6000 grains cm$^{-2}$ y$^{-1}$ and published early Holocene values rarely exceed

6000 grains cm$^{-2}$ y$^{-1}$ (but see Theuerkauf et al., 2014). Pollen diagrams from the forest steppe ecotone in European Russia are often characterized by high *Betula* percentage (Nosova et al., 2019; Shumilovskikh et al., 2018). However, there are no suitable diagrams with reliable PAR estimates from that region. It is thus difficult to judge whether high modern trap values are associated with recent land-use change or are characteristic of eastern European forests.

The comparison of regional PARs between traps and fossil estimates indicates higher fossil PAR of *Picea*, *Fagus* and *Abies*

in middle altitudes of the temperate zone (Fig. 5), which, in the case of *Abies*, represents the Europe-wide decline in *A. alba* (Tinner et al., 2013). *Picea* and *Fagus* dominate central European forests today and *Picea* is planted much beyond its natural range. However, both trees start flowering rather late in their lives and harvesting the trees at a young age may contribute to lower modern PAR values. Fossil and modern PARs for these trees in Šumava are similar, while only the highest trap values match the Holocene high values. On average there are lower modern PAR values; this may be explained by a lowering of the

treeline over the last millennia. This interpretation agrees with REVEALS reconstructions for this region, indicating a decline



in the cover abundance of *Picea* and *Fagus* (Abraham et al., 2016; Carter et al., 2018). Within a 60 km radius of the fossil sites *Picea* decreased in abundance from 70% during the Middle Holocene to 43%, compared to modern abundance. *Fagus* and *Abies* declined from Late Holocene values of 22% and 3% to currently 20% to 1% respectively (Abraham et al., 2016). The abundance of *Abies* in the Roztocze region (SE Poland; Fig. 6) provides a good analogue for the past abundance of the tree in

Šumava with maximum PAR of 1000-3000 grains cm$^{-2}$ y$^{-1}$. *Abies* disappeared from the Czech Republic during the Mediaeval Age due to forest management methods (Kozáková et al., 2011), which were not practiced in SE Poland.

## 4.2    Limitations and problems

There are significant differences between the accumulation of pollen in traps and on peatlands and lakes (Lisitsyna et al., 2011b; Pardoe et al., 2010). Differences in pollen trap design and placement in the landscape will influence the values. Trap values

are also affected by modern processes that have no impact on the fossil signal. The large consistency of the data collected in the PMP database indicates that these effects may be small. Nevertheless, some traps or individual years have unusual values and were removed from the comparison (Table S1). Despite this, the uncertainty of fossil PAR values is much greater than pollen traps, which is primarily due to the added uncertainty coming from sampling a sediment core, combined with the uncertainty of the age model (Maher, 1981). PAR from lake sediments has additional biases due to differential sedimentation

of pollen grains in lakes (Davis and Brubaker, 1973), sediment redeposition, focussing and catchment erosion (Davis et al., 1984; Pennington, 1979). Although we carefully selected the best available fossil sites PARs from lake Suminko and Rõuge Tõugjärv may be biased by lake internal processes and the addition of stream borne pollen respectively. Nevertheless, their fossil PAR estimates are in the range of values found in pollen traps. Where detailed knowledge on the sedimentation process is available the bias of sediment focussing may be reduced as in the example of Hockham Mere cited above (see also Bennett

and Buck, 2016; Bennett, 1983). Peatlands may thus seem the better choice for obtaining fossil PAR, which may be the case in northern Scandinavia (Barnekow et al., 2007; Finsinger et al., 2013), but frequent changes in the rate of peat growth lead to difficulties assessing the time represented in individual samples at many sites.

The problem of traps being contaminated by pollen brought by insects and small animals was mentioned in the method section and for this reason only Poaceae and Cyperaceae pollen were included in this comparison. However, pollen from

these two families is also often overrepresented in the pollen traps (Lisitsyna et al., 2011b), as the plants may overhang the trap opening and their pollen may fall directly into the trap. Reduced PARs in the trap may be caused by overgrowth of the vegetation or leaves temporally blocking the opening, while proximity to the forest edge would increase values compared to large open peatlands or lakes. These effects have not been systematically evaluated so far.

Detailed comparisons of vegetation data to PARs hold potential for a better understanding of the spatial representation and processes shaping the pollen signal (Matthias and Giesecke, 2014) and allow estimates of absolute pollen productivity (Sugita et al., 2009). However, forest inventory data with the detail essential for this type of study is not available for all traps. The forest cover data presented here presented here has a resolution of 1 km$^2$, so that it was insufficiently detailed to consider the

distribution of trees within hundreds of meters of the traps. Moreover, without information on standing volume or age structure the percentage cover used here is a crude measure of the vegetation producing the pollen and it is almost surprising that this





variable has some explanatory power. Forestry practices like harvesting trees that start flowering at a later age (e.g. *Picea* 30-40 years) reduce the number of trees producing pollen (Matthias et al., 2012) and bias the search for modern analogues.

## 5   Conclusions

Comparison of the mean annual PAR from traps and fossil sites showed similar ranges for *Abies*, *Alnus*, *Betula*, *Carpinus*, *Corylus*, *Fagus*, *Fraxinus*, *Picea*, *Pinus, Quercus* and *Tilia* at the continental scale. It indicates that there are no significant biases hampering the application of the PMP Database data as a modern reference to interpret the fossil record. The dataset clearly shows that climate parameters that correlate with latitude determine pollen productivity. The effect of regional forest cover is discernible, while cover within hundreds of meters is likely to be orders of magnitude stronger, but could not be

assessed here. Minimum values suggest that an 80% forest cover within 10 km of the trap results in PARs above 3200 tree pollen grains cm$^{-2}$ year$^{-1}$.

Assessment of long-distance transport indicates that threshold PAR values range from 1 to 30 grains cm$^{-2}$ y$^{-1}$ for *Carpinus*, *Corylus*, *Fagus*, *Fraxinus*, *Picea*, *Quercus* and *Tilia* may originate from beyond 200 km of a sampling site and values up to 30 grains cm$^{-2}$ y$^{-1}$ should therefore be regarded as potentially of long distance origin. The application of these threshold values

holds potential to refine and adjust reconstructions of tree distributions.

Comparison of fossil and trap datasets in individual regions with LDT threshold provided evidence for reconstruction of changes in distribution of selected species during the Holocene. Matching the target periods at individual fossil sites to trap records revealed the nearest analogous populations and provided insights into past vegetation. History of species in particular regions viewed by both ways of vegetation reconstruction show certain similarities. When ranges of trap and fossil PARs are

similar within the region, distances to the nearest analogous traps are short.

*Code availability.*   Primary trap and fossil pollen data are availible in Neotoma Palaeoecology Database https://www.neotomadb.org/. Analysis are based on the WorldClim 2 dataset of Fick and Hijmans (2017), Chorological data for the main European woody species, version 2 by Caudullo et al. (2018), European atlas of forest tree species, 2016th ed. by San-Miguel-Ayanz et al. (2016), which are availible online: http://worldclim.org/version2 , https://data.mendeley.com/datasets/hr5h2hcgg4/2 and http://www.euforgen.org/, respectively. Forest Map of Europe of Kempeneers et al. (2012) is availible on request on authors.

Code for analysis, derived data and code for figures are available in the Supplemenatry material 2.

*Author contributions.*   Design of the analysis: TG and VA. Data analysis: VA, TG and MW. Manuscript draft: VA and TG. Manuscript comments: HP, SH, CEJ, HSS, HS, SV, SH, MN, MW, SP, ST, TK, AMN and JŚM. Data collection: all authors except VA and MW.

*Competing interests.*   The authors declare no competing interests.



*Acknowledgements.* We thank to all collaborators of pollen trapping, especially to those who contributed to the PMP dataset: Pim van der Knaap, Jacqueline van Leeuwen, Karl-Dag Vorren, Heather Tinsley, Lena Barnekow, Małgorzata Latałowa, Enikő Magyari, Larissa Savelieva, Elena Pavlova, Heidi Hyyppä, Mari Kuoppamaa and Gunhild Rosqvist.





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
