# Peer review of "Patterns in recent and Holocene Pollen Accumulation Rates across Europe; the Pollen Monitoring Programme Database as a tool for vegetation reconstruction"

_Biogeosciences, 2020_

## Referee Comment (RC1) · Anonymous Referee #1 · 24 Aug 2020

The manuscript presents a new and valuable dataset that is made publicly available on a well-established database (NeotomaDB) that adheres to the World Data System and FAIR principles.

The dataset consists of pollen counts from traps located in various European regions (and associated metadata). It is potentially extremely useful to explore pollen-vegetation relationships, as has already been shown in a number of previous, regionally-focussed, studies. In comparison to these prior studies, the present manuscript explores this modern-pollen dataset at a scale that spans across much

wider geographical (latitudinal and altitudinal) gradients.

The manuscript focuses on pollen abundances of 14 pollen taxa (12 trees and shrubs genera, 2 herb families) and explores the relationships between the sum of the pollen-accumulation rates (PAR) of the 14 taxa ("total PAR") and selected environmental parameters (mean annual temperature, annual precipitation, forest cover). It also explores the relationship between tree PAR and forest cover, and the relationship between pollen deposition as a function of increasing distance to the nearest range boundary of the parent plant species. A long-distance transport threshold (LDT) is obtained that may be used to infer range-size changes based on fossil PAR values. Further, a comparison between modern and fossil PAR is presented that shows PAR-inferred population-size changes for selected taxa through time at different sites located across the latitudinal gradient.

The manuscript is at times very confusing. For instance, as far as I was able to understand, the results indicate that total PAR is strongly related with forest biomass within a 10km radius around the traps (Table S4). The text instead reports that forest cover explains 72% of the variance of total PAR. Further, Figure 3 shows how total PAR is related to latitude (besides, this relationship seems to be strongly determined by one datapoint), but does not show how total PAR are related to forest cover. Instead, the Figure shows how tree PAR is related to forest cover.

In my opinion, one of the main weaknesses of the manuscript is that the Introduction does not convey which knowledge gaps and hypotheses are being addressed. The Introduction is strongly disconnected from the Abstract, the Results, the Discussion, and the Conclusions. It dwells on how and why the Pollen Monitoring Programme was established and lists prior, regional, studies. Some concepts that are mentioned both in the Abstract and the Discussion ($CO_2$ fertilization, land use) are not found in the Introduction. Another concept (the importance of comparisons between modern and fossil pollen data, and of past and modern plant distributions and abundances) is only marginally mentioned in the Introduction, although it is important in the Discussion.

Moreover, links to Figures seem to be wrong in some places of the text, methods could be described better, tables and their captions sometimes are incomplete, and the description of the results is confusing and in some places contradictory. Further, the paragraph 3.4 "Taxa specific linkage..." seems wordy and confusing, and could be substantially shortened (besides, I was surprised when I noticed that the taxa specific linkage is placed as supplementary material for some of the taxa. Maybe it was mentioned earlier in the Mat & Methods section?). Some of the conclusions do not seem to be supported by the data.
* * *
General comments:

-> Abstract:

- the collection of [...] is important...

- statement "This dataset shows that climate parameters [...] determine pollen productivity" is not supported by the data, which shows that forest cover explains a much larger share of the variance in "total PAR".

- the statement "A signal of regional forest cover can be detected [...], while local tree cover seems more important" suggests that forest cover is substantially different from tree cover. I might have missed this difference when reading the text and suggest to better point this difference out.

- the statement "PAR values up to (smaller than?) 30 grains [...] in fossil records should therefore be interpreted as long distance transport" should be nuanced further. I suppose this refers to PAR values of the pollen taxa that were explored in this study. There are very likely some plant species whose pollen is less well dispersed or that simply produce less pollen (e.g. Larix, insect-pollinated plants). Moreover, it seems to me that the threshold value represents the PAR at 200 km from the distribution limit (as of Figure 4b). Does this mean that you (arbitrarily) consider any distance beyond 200

km as a "long distance"? If so, please state this in a clearer way in the text.

- the statement "Comparisons to fossil data from the same areas show comparable values" is unclear. What is meant exactly with the term "the same area"? Figures 6-13 show that the geographical distance between similar modern and fossil PAR values is often quite high.

- L11: "may be hard to find" seems colloquial. Replace with "do not occur, were not found in this dataset"?

- the last sentence could be replaced with a DOI link (or a NeotomaExplorer link) to the dataset in Neotoma...the link could be added on L5 after "1981 to 2017".

-> It is striking to see that you explored the relationship between the sum of the pollen-accumulation rates (PAR) of the 14 taxa ("total PAR") and forest cover (text on P8 L24-30 and Table S4). On which grounds would one expect a relationship between the PAR of herbs (Poaceae and of Cyperaceae) and forest cover?

-> A regression model for tree PAR vs forest cover is presented. The regression is based on selected tree PAR values for 3% wide forest-cover bins (Figure 3b). The regression model suggests that an 80% forest cover within 10 km radius results in tree PAR values > 3200 (Conclusions, P23 L 15). I might miss an important point, but it seems to me that the deduction is not supported by the data presented in Figure 3b. The Figure shows that values greater than 3200 tree PAR can be found even for 0% forest cover. It seems to me, instead, that tree PAR are > 20,000 for forest cover >20% (though strikingly the two sites with highest forest cover show rather low tree PAR values).

-> the manuscript shows decreasing pollen deposition as a function of increasing distance to the nearest distribution limit of the parent plant species (Figure 4). Based on this evidence, a long-distance transport threshold (LDT) for a distance of 200 km beyond the distribution limit is calculated. The thresholds could be used to infer rangesize changes based on fossil PAR values, the manuscript reports.

- While these are interesting results and a potentially useful approach, some critical discussion of this may be useful. The distribution limits were extracted from GIS shapefiles published by Caudullo et al. (2017), which are publicly available on the figshare website with associated DataCite link (Caudullo, Giovanni; Welk, Erik; San-Miguel-Ayanz, Jesús (2017): Chorological maps and data for the main European woody species. figshare. Collection. https://doi.org/10.6084/m9.figshare.c.2918528.v5). In the original manuscript where Caudullo et al. present the maps, they specifically mention that "Since the maps aim at representing the species general chorology at continental scale, providing a synthetic overview of distribution range, the mapped boundaries should not be considered as precise and sharp limits where the species is definitely present or absent, particularly at local level. Indeed, the first version of this dataset was created for the European Atlas of Forest Tree Species [16] to concisely outline the distribution ranges of described species, complementing information on the species biology and ecology. Errors and imprecision are partly inevitable, due to various causes, such as the quality of the original source, the geo-referencing procedure, the interpretation and the comparison of the sources in the same area and finally due to the limited precision of the manual digitalization process of the range borders (Fig. 1)."

- It is therefore highly questionable as to whether the distances to the distribution limits measured by Abraham et al. truly represent the actual distances to the species distribution limits. Thus, the precision and accuracy of the LDT values may be strongly overestimated and misleading. Instead of using one single distribution limit, a range of distribution limits may better represent the uncertainty of the mapped limits. Question is therefore: what LDT values would be obtained if the distribution limits of Caudullo et al. (2017) had an uncertainty? Say ca. +100km or even +200km?

- Moreover, it would be useful to show the complete data in the plots of figure 4, including the PAR values within the distribution range (thus extend the x-axes of the plots to include negative x-axis values). In theory at least, these PAR values should be greater

than PAR values around the distribution limit and beyond the limits.

-> The comparison between modern and fossil PARs is interesting (paragraphs 3.3 and 3.4).

- Paragraph 3.3. should be deeply revised and could be shortened. It could focus more on PAR-inferred presence/absence based on LDT limits that were presented previously (Figure 5), and on the identification of the closest modern counterparts of populations sizes and forest cover. Currently, some statements are descriptive and their relevance could be made clearer (for instance, on P13 L31 "Modern and fossil values agree for the sites in central Sweden at PARs between 1900-5600 grains..."). Some phrases could be removed (e.g. P13 L27 "As discussed in the main manuscript, "), other ones are unclear (e.g. P13 L25 "ignoring traps from the Caucasus and Turkey"), and several statements should be supported with references to the literature (e.g. P16 L12 "Picea abies is planted in many European regions outside its natural distribution", or "Fagus pollen occurs at fossil sites that were assumed to have never been within the distribution of the tree"), to name few examples.

- Paragraph 3.4 is enlightening. However, the term "analogue" (and "modern analogue", which is used later in the Discussion) is not approriate. With pollen records, modern analogues are generally referred to pollen assemblages (thus to vegetation composition). Here instead you refer to "comparable, similar, population size of one taxon". Using the term "modern analogue" without clarifying that you are using it with a different meaning creates confusion, particularly in the Discussion where reference is made to the early-Holocene hazel maximum.

-> There are other proxies that may be useful to determine presence of trees in fossil records (plant macrofossils and stomata). This could be mentioned in the text. Moreover, using fossil sites where such data is available could be useful to actually test the LDT limits, at least for some of the taxa.

-> Further, how do the inferences based on the LDT limits compare with inferences

made previously based on pollen percentages (or on plant macrofossils and stomata)? For instance, in a prior study (Giesecke et al., 2017 in JBiogeogr) a good agreement between the estimates of overall spread (Fig. 5a) based on different pollen percentage abundance classes was found. Some of the fossil sites were actually analysed for pollen, stomata, and plant macrofossils (e.g. Sägistalsee, Bachalpsee), but these results are not mentioned in this manuscript.

\*\*\*\*\*\*\*\*\*\*\*\*\*\*\*\*\*\*\*\*\*\*\*\*\*\*\*\*\*\*\*\*\*\*\*\*\*\*\*\*\*\*\*\*\*\*\*\*\*\*\*\*\*\*\*\*\*\*\*\*\*\*\*\*\*\*\*\*\*\*\*\*\*\*\*

Detailed comments:

- title: influx, or PAR? In the text you use PAR, not influx. Please homogenise terminology.

- Please consistently use italics for latin names.

- P6 L21-22: you extracted forest cover data from all grid cells within 10km radius of the traps. Did you then calculate a mean forest cover value? Please clarify.

P7 L8: why 271 modern samples? The abstract mentions 2742 anual samples.

P7 L10-ff.: am having trouble understanding what has been done, and why. Did you match fossil and modern PAR based on their ranking in the classes? What has been averaged? why should 500-year bins represent periods of long-term vegetation stability (fossil pollen records show fast population doubling times for some taxa)? What distance measure did you use for the cluster analysis (Euclidean distance)?

- P10 L111-12: total PAR is not shown in Figure 5.

- P11 L12: here the conclusion is presented before the data and then the data are presented to support the initially stated conclusion. Please reverse the line of arguments.

- P11 L13: am having trouble to understand why 92 pairs were obtained. please clarify.

- P11 L14: cannot find t-test and p-values on figure 6. Neither were "t-test" and "pvalues" mentioned in the Material and Methods section. Please clarify why and what has been done.

P20 L12: MAT has a very low influence on pollen deposition (see Table S4).

P20 L17-18: Which data/analysis/result shows that biomass cannot explain the latitudinal PAR gradient?

P 20 L31: why cannot the details be discussed here?

P21 L14: statement "modern PAR for Betula and Pinus are not found in fossil samples" contradicts a prior statement (P10 L12 "Maximum PAR in traps are always higher compared to fossil situations, with the exception of Corylus").

P22 L16: the effects are possibly small in relation to the wide environmental gradient.

P22 L34-35: these are interesting notions. Please mention the usefulness of modern PARs (as listed here), and their importance for ecological and biogeographical studies, in the Introduction.

P22 last line: remove "and it is almost surprising that...explanatory power". It might be surprising, but is a fact that seems to contradict a prior study (Matthias et al.). Could the gradient length be an important factor here?

-> Figures 6-13: - the a) and b) frames could be merged by using horizontal boxplots (instead of barplots) in a), and adding b) as an overlay;

- font sizes are too small;

-> References: - add: Caudullo, G., Welk, E., San-Miguel-Ayanz, J., 2017. Chorological maps for the main European woody species. Data in Brief 12, 662–666. https://doi.org/10.1016/j.dib.2017.05.007

-> In Table S4:

- caption: what does "alternatively" mean here? Please clarify.

- Forest biomass, or forest cover (as of P6 L20-21)?

- are the "Adjusted R2" values of the "adjusted PAR" values the ones obtained with the Andersen correction factors (P6 L23-24)? Please clarify.

- please add r2 values for "tree PAR" (not total PAR) vs forest cover.

---

## Referee Comment (RC2) · Anonymous Referee #2 · 2 Sep 2020

Note to editor: The line numbers in the manuscript appear to be sometimes inconsistent. I assume this is a fault with the submission process not the authors' fault. Apologies if any confusion arises from this.

General comments

An interesting manuscript with great potential to improve interpretations of fossil pollen records. It is clear that a lot of careful thought and work has gone into this study. While it has potential to be very useful, the structure and clarity of the work could be

improved. I think in particular the structure could be refined to be more consistent throughout, and in the Introduction and Discussion sections, sub-headings introduced to clarify the development of the argument.

There seem to be various strands to this paper:

1. The correlations of PAR with parameters such as forest cover and temperature as calculated from the modern samples

2. The question of long distance transportation beyond the current extent of the parent plant taxa

3. The relationship between modern PAR and fossil PAR for selected taxa

I feel that if the first two are clearly and separately addressed, it would be easier for the authors to address the third point coherently. At the moment, the results broadly follow this structure, but the Introduction and Discussion do not, I suspect if they did, the paper would flow better.

Specific points by section

Introduction

I was a little surprised that so much of the introduction dealt with history. Although I think this would merit its own sub-section, I think it would be better to focus on scientific questions addressed in the manuscript. Why pollen traps are an appropriate analogue for fossil records should be introduced. It would be nice to see some mention of species ranges and their possible fluctuation over time, and whether present day species distributions can be considered to be in equilibrium with climate. Another factor that could be addressed in the introduction is pollen dispersal and deposition; how far does pollen usually travel? This would set up the argument for your chosen LDT threshold.

2 Methods

General comment: all botanical names including species epithets need to be written

with their authorities the first time they are mentioned in the manuscript. Up to date authorities can be found here http://www.worldfloraonline.org/. Upon first use, a species must be written out in full even if its genus has been mentioned by name previously. This is to avoid confusion between genera that start with the same letter. So for instance, Pinus sylvestris L. could be shortened to P. sylvestris, but then Pinus mugo Turra needs to be written as such before it can be abbreviated to P. mugo

Figure 1: As there is so much overlap on the map between modern and fossil sites, I think separating this out into two side by side maps with one showing fossil and one modern samples would make it clearer, and would also make it easier to go back and check locations of fossil sites as I was trying to do so later in the manuscript.

On page 7, 2742 samples were mentioned as being in the database (Section 3.1) but on line 8 only 271 are mentioned- which number was in the analysis?

2.2 Data Collection

It would be interesting to see a plot of surface area of trap against PAR to test for a relationship there and potentially be able to correct if one exists.

2.3 Investigated taxa and Environmental parameter

Why was 200km chosen as the threshold for LDT?

2.4 Comparison

Page 12: Figure 5 is a bit tricky to interpret, however, once I had realised what it was supposed to be showing I saw its value. I particularly like the LDT cut-off, which will be potentially very useful in interpreting fossil records. I was surprised, however, that LDTs did not receive further attention in the discussion section as it seems that they are a tangible, useful output from the work.

3.2 Recent and fossil PAR

Figures 6 onwards: In caption, specify that distribution of taxon is in grey. These taxonspecific figures are in general, I think, quite useful. I hope they are reproduced a little bigger (at least $\frac{1}{2}$ a page each) so that the details on the maps are easily readable. If this is not possible, maps should perhaps be split into separate figures to improve legibility. I am not sure the multi-coloured coding for the PAR values adds much to the figure- you could probably do away with (b) and still retain the meaning of the figures. I am also not clear on why, in the fossil graphs, only certain colours are included- why is only the highest PAR of interest?

3.4 Taxa specific results

How were the 'main taxa' to be presented chosen for this section? It seems odd that some are arbitrarily in the SM, particularly arboreal pollen which was presented in Figure 2.

Figures need to be referred to consistently in this section.

3.4.2 Betula

Why are traps from the Caucasus and Turkey ignored?

4.1 Discussion

The first sentence of the discussion doesn't seem to tally with the results- it looks, from your data, like the relationship between modern and fossil PAR is actually quite complex and variable. I don't necessarily think this is a bad thing, however; the paper presents a quantitative dataset that could potentially be used to help researchers quantify what their PARs from fossil data actually mean.

Line 22 onward: This paragraph seems to be about LDT, but that isn't explicitly stated.

Last paragraph of page 20 onwards appears to be, broadly, taxon-specific discussion of modern and fossil PARs; it would be better if this were clearly signposted and possibly split up with sub-headings.

I would be interested to see some consideration of how these results might be useful in feeding into quantitative reconstructions of vegetation. Although PDD models tend to deal in percentages, surely this approach (on cores with appropriate chronologies) could open the door to future models being calibrated using PARs, an interesting prospect for vegetation reconstructions, particularly given your LDT estimates.

4.2 Limitations

Line 21: Why are only 3 fossil sites listed here? Are the others not likely to show any bias?

Line 28: Unclear which analysis only included Poaceae and Cyperaceae

5 Conclusions

This is succinct and mostly well-structured but would benefit from a closing statement outlining the applications and take home message of the paper.

Technical comments

Line 12: replace 'case' with 'cause'

Line 13: Could be rephrased as 'hopefully serves to improve interpretations' (or remove hopefully- I think it definitely will).

Line 1: A good recent reference here would be Haselhorst 2020 (DOI: 10.1111/jvs.12897) showing high interannual variability in the tropics too- strengthens general argument.

Line 7: remove comma after 'Although'.

Line 13: something strange happening here after 'values variation'- typo?

Line 19: The sentences about fossil pollen seem out of place here as this section of the results is regarding modern pollen.

Line 5: 'main text' – where in the manuscript is being referred to?

Line 31 paragraph: I think this paragraph might be better placed at the end of this section. The phrasing in Line 32 seems a little odd as you go on to give an example of linking fossil to modern PARs- perhaps delete the 'details cannot be discussed' sentence.

3.4.8 Line ? (Line numbers unclear) correct LTD to LDT and remove 'threshold'

---

## Author Comment (AC1) · 29 Sep 2020

We are pleased that both reviewers see the value of the database that we aim to make publically available with this publication. We also appreciate the comments on the manuscript that will certainly help to improve it. We like to take this opportunity to respond to some of the comments, each response is introduced by ### and finished by $$$

Anonymous Referee #1 The manuscript presents a new and valuable dataset that is

made publicly available on a well-established database (NeotomaDB) that adheres to the World Data System and FAIR principles. The dataset consists of pollen counts from traps located in various European regions (and associated metadata). It is potentially extremely useful to explore pollen-vegetation relationships, as has already been shown in a number of previous, regionally-focussed, studies. In comparison to these prior studies, the present manuscript explores this modern-pollen dataset at a scale that spans across much wider geographical (latitudinal and altitudinal) gradients. The manuscript focuses on pollen abundances of 14 pollen taxa (12 trees and shrubs genera, 2 herb families) and explores the relationships between the sum of the pollen-accumulation rates (PAR) of the 14 taxa ("total PAR") and selected environmental parameters (mean annual temperature, annual precipitation, forest cover). It also explores the relationship between tree PAR and forest cover, and the relationship between pollen deposition as a function of increasing distance to the nearest range boundary of the parent plant species. A long-distance transport threshold (LDT) is obtained that may be used to infer range-size changes based on fossil PAR values. Further a comparison between modern and fossil PAR is presented that shows PAR-inferred population-size changes for selected taxa through time at different sites located across the latitudinal gradient.

The manuscript is at times very confusing. For instance, as far as I was able to understand, the results indicate that total PAR is strongly related with forest biomass within a 10km radius around the traps (Table S4). The text instead reports that forest cover explains 72% of the variance of total PAR.

**We realize that due to the many analyses conducted and co-authors contributions some arguments were unresolved and miscommunications occurred. Unfortunately, we were not able to obtain biomass data for European vegetation cover and used "forest cover" instead. We will work to streamline the manuscript correctly indicating the usage of "forest cover" throughout. $$$**

Further, Figure 3 shows how total PAR is related to latitude (besides, this relationship

seems to be strongly determined by one datapoint),

**We tested the influence of the data point on Spitsbergen, which of cause has a strong pull, but removing it is not changing our observations and conclusions as the following table with that point removed will show: PAR adjusted_PAR latitude 0.09 0.37 MAT 0.19 0.34 Forest cover 10 km 0.69 0.79 latitude+MAT+Forest cover 10 km 0.74 0.78 latitude+MAT+Forest cover 10 km+elevation 0.70 0.80**

We would add this table with and without the point on Spitsbergen to the supplementary information of the manuscript. $$$

but does not show how total PAR are related to forest cover. Instead, the Figure shows how tree PAR is related to forest cover. -> It is striking to see that you explored the relationship between the sum of the pollen-accumulation rates (PAR) of the 14 taxa ("total PAR") and forest cover (text on P8L24-30 and Table S4). On which grounds would one expect a relationship between the PAR of herbs (Poaceae and of Cyperaceae) and forest cover?

**We cannot expect a strong relationship between total PAR and forest cover as there may be 100% cover in a forest with very low productivity in northern Finland or a high productivity in southern Europe. By broadly accounting for differences in pollen productivity between trees and also grasses we are able to investigate a relationship between adjusted PAR and latitude and mean annual temperature. In a multiple regression model forest cover is however contributing to explain the variance in PAR and adjusted PAR. We illustrate in Fig. 3B that there is however a relationship between the minimum PAR values and the amount of forest cover. This relationship is important for interpretations as we are often interested to provide conservative interpretations on this "minimum" side of the spectrum e.g. during the initial spread of forest after the ice age. $$$**

In my opinion, one of the main weaknesses of the manuscript is that the Introduction does not convey which knowledge gaps and hypotheses are being addressed. The

Introduction is strongly disconnected from the Abstract, the Results, the Discussion, and the Conclusions. It dwells on how and why the Pollen Monitoring Programme was established and lists prior, regional, studies. Some concepts that are mentioned both in the Abstract and the Discussion ($CO_2$ fertilization, land use) are not found in the Introduction. Another concept (the importance of comparisons between modern and fossil pollen data, and of past and modern plant distributions and abundances) is only marginally mentioned in the Introduction, although it is important in the Discussion.

**We acknowledge that that the introduction need to be improved addressing open questions. However, this manuscript also aims to introduce the database itself, which is a result of decades of research with its own history and like to find room for this aspect as well. $$$**

(besides, I was surprised when I noticed that the taxa specific linkage is placed as supplementary material for some of the taxa. Maybe it was mentioned earlier in the Mat & Methods section?).

**We selected the 8 most important taxa to be included in the main text. The remaining taxa including Cyperaceae and Poaceae, for which we assume the trapping data to be potentially biased are placed in the supplementary. We inform the reader about this in the Methods section. $$$**

- the statement "A signal of regional forest cover can be detected [...], while local tree cover seems more important" suggests that forest cover is substantially different from tree cover. I might have missed this difference when reading the text and suggest to better point this difference out.

**We use only data from 10 km radius. We use forest cover in whole manuscript. Sentence merged with previous sentence. $$$**

- the statement "Comparisons to fossil data from the same areas show comparable values" is unclear. What is meant exactly with the term "the same area"? Figures 6-13

show that the geographical distance between similar modern and fossil PAR values is often quite high.

**Yes, this was unclear, because in Figures 6-13 we link individual traps or trap areas. In this sentence, we mean the comparison at level of same trap region (Fig. 5). $$$**

-> A regression model for tree PAR vs forest cover is presented. The regression is based on selected tree PAR values for 3% wide forest-cover bins (Figure 3b). The regression model suggests that an 80% forest cover within 10 km radius results in tree PAR values > 3200 (Conclusions, P23 L 15). I might miss an important point, but it seems to me that the deduction is not supported by the data presented in Figure3b. The Figure shows that values greater than 3200 tree PAR can be found even for 0% forest cover. It seems to me, instead, that tree PAR are > 20,000 for forest cover >20% (though strikingly the two sites with highest forest cover show rather low tree PAR values).

**We believe that there is value in looking at the minimum values as clarified above. So yes it is well possible to obtain much higher PAR values at 80% forest cover, but values lower than 3200 grains per cm2 are unlikely. This relationship is important for interpretations as we are often interested to provide conservative interpretations on this "minimum" side of the spectrum e.g. during the initial spread of forest after the ice age. $$$**

-> the manuscript shows decreasing pollen deposition as a function of increasing distance to the nearest distribution limit of the parent plant species (Figure 4). Based on this evidence, a long-distance transport threshold (LDT) for a distance of 200 km beyond the distribution limit is calculated. The thresholds could be used to infer range-size changes based on fossil PAR values, the manuscript reports. - While these are interesting results and a potentially useful approach, some critical discussion of this may be useful. The distribution limits were extracted from GIS shapefiles published

by Caudullo et al. (2017), which are publicly available on the figshare website with associated DataCite link (Caudullo, Giovanni; Welk, Erik; San-Miguel-Ayanz, Jesús (2017): Chorological maps and data for the main European woody species. figshare. Collection. https://doi.org/10.6084/m9.figshare.c.2918528.v5).

**We cite the version we used - v2. $$$**

In the original manuscript where Caudullo et al. present the maps, they specifically mention that "Since the maps aim at representing the species general chorology at continental scale, providing a synthetic overview of distribution range, the mapped boundaries should not be considered as precise and sharp limits where the species is definitely present or absent, particularly at local level. Indeed, the first version of this dataset was created for the European Atlas of Forest Tree Species [16] to concisely outline the distribution ranges of described species, complementing information on the species biology and ecology. Errors and imprecision are partly inevitable, due to various causes, such as the quality of the original source, the geo-referencing procedure, the interpretation and the comparison of the sources in the same area and finally due to the limited precision of the manual digitalization process of the range borders (Fig. 1)."

**We are aware that these distribution maps have uncertainties. However, PARs are unlikely to be much effected by the occurrence of parent trees at very low abundance. Due to plantations we did not look at the western distribution limits, but only explore latitudinal limits of tree distributions. We agree that additional critical discussion of these maps will be useful. $$$**

- It is therefore highly questionable as to whether the distances to the distribution limits measured by Abraham et al. truly represent the actual distances to the species distribution limits. Thus, the precision and accuracy of the LDT values may be strongly overestimated and misleading. Instead of using one single distribution limit, a range of distribution limits may better represent the uncertainty of the mapped limits. Question

is therefore: what LDT values would be obtained if the distribution limits of Caudullo et al. (2017) had an uncertainty? Say ca. +100km or even +200km?

**Since we focus in these comparisons on the northern distribution limits we believe that rare occurrences of trees will be less and less likely as further north we go. So a given uncertainty in the maps would not change the relative differences between taxa nor the order of magnitude of the absolute values, which in any case are gross estimates. Still these numbers are useful as modern analogues and represent guidelines rather than hard thresholds. $$$**

Moreover, it would be useful to show the complete data in the plots of figure 4, including the PAR values within the distribution range (thus extend the x-axes of the plots to include negative x-axis values). In theory at least, these PAR values should be greater than PAR values around the distribution limit and beyond the limits.

**We included the data for all traps in Fig. 4a). Within the area of distribution area the data is represented as boxplots outside as a dots. $$$**

-> The comparison between modern and fossil PARs is interesting (paragraphs 3.3 and3.4). - Paragraph 3.3. should be deeply revised and could be shortened. It could focus more on PAR-inferred presence/absence based on LDT limits that were presented previously (Figure 5), and on the identification of the closest modern counterparts of populations sizes and forest cover. Currently, some statements are descriptive and their relevance could be made clearer (for instance, on P13 L31 "Modern and fossil values agree for the sites in central Sweden at PARs between 1900-5600 grains..."). Some phrases could be removed (e.g. P13 L27 "As discussed in the main manuscript, "), other ones are unclear (e.g. P13 L25 "ignoring traps from the Caucasus and Turkey"), and several statements should be supported with references to the literature (e.g. P16 L12"Picea abies is planted in many European regions outside its natural distribution", or" Fagus pollen occurs at fossil sites that were assumed to have never been within the distribution of the tree"), to name few examples.

**Paragraph 3.3 will be shortened to the first and two last sentences. The rest of summarizing sentences we removed. $$$**

- Paragraph 3.4 is enlightening. However, the term "analogue" (and "modern analogue", which is used later in the Discussion) is not appropriate. With pollen records, modern analogues are generally referred to pollen assemblages (thus to vegetation composition). Here instead you refer to "comparable, similar, population size of one taxon". Using the term "modern analogue" without clarifying that you are using it with a different meaning creates confusion, particularly in the Discussion where reference is made to the early-Holocene hazel maximum.

**We agree that the term "modern analogue" evokes comparison of assemblages of pollen percentages (sensu Overpeck et al., 1985), however we, similarly to Overpeck et al. (1985), think that "modern analogue" should not be reduced to comparisons of modern pollen assemblages, but describes any link between a modern situation and its resemblance of a fossil find, including PAR. Also in geology and macro-ecology the term "modern analogue" is used in a broader context as exemplified by these publications: Sidder, A. (2020), Ancient sea levels in South Africa may offer modern analogues, Eos, 101, https://doi.org/10.1029/2020EO147001. ; Horsák, M., Chytrá¡, M., Hájková, P., Hájek, M., Danihelka, J., Horsáková, V., Ermakov, N., German, D. A., Kočí, M., Lustyk, P., Nekola, J. C., Preislerová, Z. and Valachovič, M.: European glacial relict snails and plants: environmental context of their modern refugial occurrence in southern Siberia, Boreas, 44(4), 638–657, doi:10.1111/bor.12133, 2015. We did look for a different terminology but could not find a better term for what we are comparing and are happy to clarify this in the introduction. $$$**

-> There are other proxies that may be useful to determine presence of trees in fossil records (plant macrofossils and stomata). This could be mentioned in the text. Moreover, using fossil sites where such data is available could be useful to actually test the LDT limits, at least for some of the taxa. -> Further, how do the inferences based on the LDT limits compare with inferences made previously based on pollen percentages

(or on plant macrofossils and stomata)? For instance, in a prior study (Giesecke et al., 2017 in JBiogeogr) a good agreement between the estimates of overall spread (Fig. 5a) based on different pollen percentage abundance classes was found. Some of the fossil sites were actually analysed for pollen, stomata, and plant macrofossils (e.g. Sägistalsee, Bachalpsee), but these results are not mentioned in this manuscript.

**This is an interesting aspect, which indeed we did not address in this manuscript. In making the database accessible we hope that questions like these will lead to additional usages of the database. We would be interested to explore it but feel it would add too much extra analysis and text and to the current manuscript. $$$**

Detailed comments: - P7 L8: why 271 modern samples? The abstract mentions 2742 annual samples.

**While the database contains all data that was submitted, we only considered trap record with at least 3 years and thus obtained 271 traps that we base the analysis on. $$$**

- P11 L13: am having trouble to understand why 92 pairs were obtained. please clarify.

**We have 15 taxa in 7 regions, which make 105 cases that we investigate, however some regions lack species in the fossil and/or trap record and thus it is only possible to perform the t-test for 92 pairs of trap and fossil sites. $$$**

- P11 L14: cannot find t-test and p-values on figure 6. Neither were "t-test" and "p-values" mentioned in the Material and Methods section. Please clarify why and what has been done.

**We are sorry for this omission and will add it to the Methods section and add a table with p-values to the Supplementary. $$$**

-> Figures 6-13: - the a) and b) frames could be merged by using horizontal boxplots (instead of barplots) in a), and adding b) as an overlay;

**I have tried to use horizontal boxplots (Fig. R1), but the highest values enlarge the x-axis while the lowest classes are not visible. The way of our presentation allows to appreciate the general pattern of mean values for trap areas (a) and we make the full variability of individual traps visible within the trap areas by coloured squares (b). - font sizes are too small; Font was enlarged to 2.5 for names of the fossil sites/trap areas and 2 for rest of the text. $$$**

[Figure]

*Abies*

**Fig. 1.** Fig. R1: Figure 6 - current version (first two panels) and proposed version (third panel, boxplot).

---

## Author Comment (AC2) · 29 Sep 2020

We are pleased that both reviewers see the value of the database that we aim to make publically available with this publication. We also appreciate the comments on the manuscript that will certainly help to improve it. We like to take this opportunity to respond to some of the comments, each response is introduced by ### and finished by $$$

Anonymous Referee #2 I feel that if the first two are clearly and separately addressed,

it would be easier for the authors to address the third point coherently. At the moment, the results broadly follow this structure, but the Introduction and Discussion do not, I suspect if they did, the paper would flow better.

**We are grateful for this suggestion and will work to implement it in a revised manuscript. $$$**

Specific points by section Introduction I was a little surprised that so much of the introduction dealt with history. Although I think this would merit its own sub-section, I think it would be better to focus on scientific questions addressed in the manuscript. Why pollen traps are an appropriate analogue for fossil records should be introduced. It would be nice to see some mention of species ranges and their possible fluctuation over time, and whether present day species distributions can be considered to be in equilibrium with climate. Another factor that could be addressed in the introduction is pollen dispersal and deposition; how far does pollen usually travel? This would set up the argument for your chosen LDT threshold.

**We appreciate this suggestion to improve the organisation of the manuscript and will move the historical aspects of the development of the network (see response to rev1) to a subsection. $$$**

2 Methods General comment: all botanical names including species epithets need to be written with their authorities the first time they are mentioned in the manuscript. Up to date authorities can be found here http://www.worldfloraonline.org/. Upon first use, a species must be written out in full even if its genus has been mentioned by name previously. This is to avoid confusion between genera that start with the same letter. So for in-stance, Pinus sylvestris L. could be shortened to P. sylvestris, but then Pinus mugo Turra needs to be written as such before it can be abbreviated to P. mugo

**We can add authorities to the botanical names, however Biogeosciences is not taxonomical journal and many papers state species without authorities. $$$**

Figure 1: As there is so much overlap on the map between modern and fossil sites, I think separating this out into two side by side maps with one showing fossil and one modern samples would make it clearer, and would also make it easier to go back and check locations of fossil sites as I was trying to do so later in the manuscript. ### Thank you for this comment. We already separated a map of only fossil sites and it is better readable. $$$

On page 7, 2742 samples were mentioned as being in the database (Section 3.1) button line 8 only 271 are mentioned- which number was in the analysis?

**We added to Section 3.1: "Considering the trap record with 3 years and more we obtained 271 mean trap assemblages." $$$**

2.2 Data Collection It would be interesting to see a plot of surface area of trap against PAR to test for a relationship there and potentially be able to correct if one exists.

**This is an interesting point and the reviewer is invited to try this out as all this information will be available. Do you have any idea which mechanism can stay beyond? $$$**

2.3 Investigated taxa and Environmental parameter Why was 200km chosen as the threshold for LDT?

**Regional pollen is assumed to correspond to the vegetation cover in 100 by 100 km (Hellmann et al 2008) so doubling that distance represents a good rational. Additionally, we considered the uncertainty of the maps. $$$**

2.4 Comparison Page 12: Figure 5 is a bit tricky to interpret, however, once I had realised what it was supposed to be showing I saw its value. I particularly like the LDT cut-off, which will be potentially very useful in interpreting fossil records. I was surprised, however, that LDTs did not receive further attention in the discussion section as it seems that they are a tangible, useful output from the work.

**We are happy to develop the topic in the discussion. $$$**

3.2 Recent and fossil PAR Figures 6 onwards: In caption, specify that distribution of taxon is in grey. These taxon-specific figures are in general, I think, quite useful. I hope they are reproduced a little bigger (at least a page each) so that the details on the maps are easily readable. If this is not possible, maps should perhaps be split into separate figures to improve legibility. I am not sure the multi-coloured coding for the PAR values adds much to the figure- you could probably do away with (b) and still retain the meaning of the figures. I am also not clear on why, in the fossil graphs, only certain colours are included - why is only the highest PAR of interest?

**Added: "... distribution of the species (gray, Caudullo et al., 2018 ...".**

Unfortunately, one page per figure would surpass our APC budget.

We will improve legibility by abbreviating names of fossil sites, so we make use of blank space and the map is larger now.

Multi-coloured coding helps to link the fossil and trap PAR values of the same height/class. In the fossil stratigraphic plot we point out the class of interest, coloured squares in 6b) illustrate the variability of traps within one trap area.

Boxplot would be one option as proposed by reviewer 1 (see response there), black and white bar-plots for the fossil and trap record without any colours would not visualize the analysis we did.

Only certain classes appeared in the fossil record. We picked the highest PAR class from the fossil record, because it represents also the densest population of the source plants. $$$

3.4 Taxa specific results How were the 'main taxa' to be presented chosen for this section? It seems odd that some are arbitrarily in the SM, particularly arboreal pollen which was presented in Figure 2. Figures need to be referred to consistently in this section.

**We do not want to flood reader by all species. Fossil-trap links of species selected**

for the main text show nicely changes of distribution patterns, whereas species in the SM can suffer from certain biases and limitations (See Cyperaceae and Poaceae in section 4.2) $$$

3.4.2 Betula Why are traps from the Caucasus and Turkey ignored?

**Reworded to "Letting aside traps from the Caucasus and Turkey...(, because those two areas have different species Betula than rest of the trap areas)" $$$**

4.1 Discussion The first sentence of the discussion doesn't seem to tally with the results- it looks, from your data, like the relationship between modern and fossil PAR is actually quite complex and variable. I don't necessarily think this is a bad thing, however; the paper presents a quantitative dataset that could potentially be used to help researchers quantify what their PARs from fossil data actually mean.

**We included mention about complexity of the relationship to the first sentence. $$$**

I would be interested to see some consideration of how these results might be useful in feeding into quantitative reconstructions of vegetation. Although PDD models tend to deal in percentages, surely this approach (on cores with appropriate chronologies) could open the door to future models being calibrated using PARs, an interesting prospect for vegetation reconstructions, particularly given your LDT estimates.

**We included in the discussion possibilities of PAR vs. PDD and wider use of LTD estimates in quantitative reconstruction. $$$**

4.2 Limitations Line 21: Why are only 3 fossil sites listed here? Are the others not likely to show any bias?

**The fossil PAR values at the 3 sites are rather high so we suspect that lake internal procceses may explain these values. In generall the reviwer is right these biases may also occure at other sites and we will add this to the discussion in this section. $$$**

---

## Author Response (AR1)

We are pleased that both reviewers see the value of the database that we aim to make
 publically available with this publication. We also appreciate the comments on the manuscript
 that will certainly help to improve it. We like to take this opportunity to respond to some of

4 the comments.

5 6

- 7 Anonymous Referee #1
- 8 The manuscript presents a new and valuable dataset that is made publicly available on a well-
- 9 established database (NeotomaDB) that adheres to the World Data System and FAIR
- 10 principles. The dataset consists of pollen counts from traps located in various European
- 11 regions (and associated metadata). It is potentially extremely useful to explore pollen-12 vegetation relationships, as has already been shown in a number of previous, regionally-
- vegetation relationships, as has already been shown in a number of previous, regionally focussed, studies. In comparison to these prior studies, the present manuscript explores this
- 14 modern-pollen dataset at a scale that spans across much wider geographical (latitudinal and
- 15 altitudinal) gradients. The manuscript focuses on pollen abundances of 14 pollen taxa (12
- 16 trees and shrubs genera, 2 herb families) and explores the relationships between the sum of
- the pollen-accumulation rates (PAR) of the 14 taxa ("total PAR") and selected environmental
- 18 parameters (mean annual temperature, annual precipitation, forest cover). It also explores the
- relationship between tree PAR and forest cover, and the relationship between pollen
- 20 deposition as a function of increasing distance to the nearest range boundary of the parent
- 21 plant species. A long-distance transport threshold (LDT) is obtained that may be used to infer
- 22 range-size changes based on fossil PAR values. Further a comparison between modern and
- 23 fossil PAR is presented that shows PAR-inferred population-size changes for selected taxa
- through time at different sites located across the latitudinal gradient.
- 25
- The manuscript is at times very confusing. For instance, as far as I was able to under-stand, the results indicate that total PAR is strongly related with forest biomass within a 10km radius
- around the traps (Table S4). The text instead reports that forest cover explains 72% of the
- 29 variance of total PAR.
- 30 We realize that due to the many analyses conducted and co-authors contributions some
- arguments were unresolved and miscommunications occurred. Unfortunately, we were not
- 32 able to obtain biomass data for European vegetation cover and used "forest cover" instead.
- 33 We will work to streamline the manuscript correctly indicating the usage of "forest cover"
- throughout. 72% of explained variance was error in the analysis, Table S4 for updated results.
- 35

Further, Figure 3 shows how total PAR is related to latitude (besides, this relationship seems to be strongly determined by one datapoint),

We tested the influence of the data point on Spitsbergen, which of cause has a strong pull, but

removing it is not changing our observations and conclusions as the following table with that

40 point removed will show:

|                                 | PAR  | adjusted_PAR |
|---------------------------------|------|--------------|
| latitude                        | 0.09 | 0.22         |
| MAT                             | 0.19 | 0.26         |
| Forest cover 10 km              | 0.18 | 0.16         |
| latitude+MAT+Forest cover 10 km | 0.35 | 0.42         |
| latitude+MAT+Forest cover 10    |      |              |
| km+elevation                    | 0.49 | 0.54         |
|                                 |      |              |

41

42 We would add this table with and without the point on Spitsbergen to the supplementary

43 information of the manuscript.

**44**

but does not show how total PAR are related to forest cover. Instead, the Figure shows howtree PAR is related to forest cover.

- 47 -> It is striking to see that you explored the relationship between the sum of the pollen-
- 48 accumulation rates (PAR) of the 14 taxa ("total PAR") and forest cover (text on P8L24-30
- 49 and Table S4). On which grounds would one expect a relationship between the PAR of herbs
- 50 (Poaceae and of Cyperaceae) and forest cover?
- 51

We cannot expect a strong relationship between total PAR and forest cover as there may be 52 100% cover in a forest with very low productivity in northern Finland or a high productivity 53 in southern Europe. By broadly accounting for differences in pollen productivity between 54 trees and also grasses we are able to investigate a relationship between adjusted PAR and 55 56 latitude and mean annual temperature. In a multiple regression model forest cover is however contributing to explain the variance in PAR and adjusted PAR. We illustrate in Fig. 3B that 57 there is however a relationship between the minimum PAR values and the amount of forest 58 59 cover. This relationship is important for interpretations as we are often interested to provide conservative interpretations on this "minimum" side of the spectrum e.g. during the initial 60 spread of forest after the ice age. 61

62

In my opinion, one of the main weaknesses of the manuscript is that the Introduction does not
convey which knowledge gaps and hypotheses are being addressed. The Introduction is
strongly disconnected from the Abstract, the Results, the Discussion, and the Conclusions. It

- 66 dwells on how and why the Pollen Monitoring Programme was established and lists prior,
- regional, studies. Some concepts that are mentioned both in the Abstract and the Discussion
- 68 (CO2 fertilization, land use) are not found in the Introduction. Another concept (the
- 69 importance of comparisons between modern and fossil pollen data, and of past and modern
- 70 plant distributions and abundances) is only marginally mentioned in the Introduction,
- 71 although it is important in the Discussion.

72 Introduction was improved addressing open questions (CO2 fertilization, land use, MAT).

- However, this manuscript also aims to introduce the database itself, which is a result of
- 74 decades of research with its own history and thus development PMP network has a special
- 75 subsection in the Introduction.
- 76 77

Moreover, links to Figures seem to be wrong in some places of the text, methods could bedescribed better,

- tables and their captions sometimes are incomplete, and the description of the results is
- confusing and in some places contradictory. Further, the paragraph 3.4 "Taxa specific
- 82 linkage..." seems wordy and confusing, and could be substantially shortened
- 83 We attempted to place the figure close to the first mention in the text, automatic knitting
- 84 process of the MarkDown document might move them one Figure per page. In a later
- comment you state the opposite: "The comparison between modern and fossil PARs is
- interesting (paragraphs 3.3 and 3.4)". Assuming "Paragraph 3.4 is enlightening" we agree that
  sect. 3.3 can be shortened.
- 88 89
- 90 (besides, I was surprised when I noticed that the taxa specific linkage is placed as
- 91 supplementary material for some of the taxa. Maybe it was mentioned earlier in the Mat & Matheda spatian?)
- 92 Methods section?).

| 93  | We selected the 8 most important taxa to be included in the main text. The remaining taxa        |
|-----|--------------------------------------------------------------------------------------------------|
| 94  | including Cyperaceae and Poaceae, for which we assume the trapping data to be potentially        |
| 95  | biased are placed in the supplementary. We inform the reader about this in the Methods and       |
| 96  | Results section.                                                                                 |
| 97  |                                                                                                  |
| 98  |                                                                                                  |
| 99  | Some of the conclusions do not seem to be supported by the data                                  |
| 100 | Conclusions were shortened and they are supported by the data                                    |
| 101 | conclusions were shortened and mey are supported by the data.                                    |
| 101 |                                                                                                  |
| 102 |                                                                                                  |
| 103 | General comments:                                                                                |
| 104 | -> Abstract:                                                                                     |
| 105 | - the collection of [] is important                                                              |
| 106 | Changed                                                                                          |
| 107 |                                                                                                  |
| 108 | - statement "This dataset shows that climate parameters [] determine pollen productivity" is     |
| 109 | not supported by the data, which shows that forest cover explains a much larger share of the     |
| 110 | variance in "total PAR".                                                                         |
| 111 | Rephrased in the sense that total PAR is determined by forest cover.                             |
| 112 |                                                                                                  |
| 113 | - the statement "A signal of regional forest cover can be detected [], while local tree cover    |
| 114 | seems more important" suggests that forest cover is substantially different from tree cover. I   |
| 115 | might have missed this difference when reading the text and suggest to better point this         |
| 116 | difference out.                                                                                  |
| 117 | We use only data from 10 km radius. We use forest cover in whole manuscript. Sentence            |
| 118 | merged with previous sentence. Tree cover deleted.                                               |
| 119 |                                                                                                  |
| 120 | - the statement "PAR values up to (smaller than?) 30 grains [] in fossil records should          |
| 121 | therefore be interpreted as long distance transport" should be nuanced further. I suppose this   |
| 122 | refers to PAR values of the pollen taxa that were explored in this study. There are very likely  |
| 123 | some plant species whose pollen is less well dispersed or that simply produce less pollen (e.g.  |
| 124 | Larix, insect-pollinated plants). Moreover, it seems to me that the threshold value represents   |
| 125 | the PAR at 200 km from the distribution limit (as of Figure 4b). Does this mean that you         |
| 126 | (arbitrarily) consider any distance beyond 200 km as a "long distance"? If so, please state this |
| 127 | in a clearer way in the text.                                                                    |
| 128 | Statement was specified by taxa used in the paper and the distance 200 km.                       |
| 129 |                                                                                                  |
| 130 | - the statement "Comparisons to fossil data from the same areas show comparable values" is       |
| 131 | unclear. What is meant exactly with the term "the same area"? Figures 6-13 show that the         |
| 132 | geographical distance between similar modern and fossil PAR values is often quite high.          |
| 133 | Yes, this is unclear, because in Figures 6-13 we link individual traps or trap areas. In this    |
| 134 | sentence, we mean the comparison at level of same trap region (Fig. 5).                          |
| 135 |                                                                                                  |
| 136 | - L11: "may be hard to find" seems colloquial. Replace with "do not occur, were not found in     |
| 137 | this dataset"?                                                                                   |
| 138 | Replaced                                                                                         |
| 139 |                                                                                                  |
| 140 | - the last sentence could be replaced with a DOI link (or a Neotoma Explorer link) to the        |
| 141 | dataset in Neotomathe link could be added on L5 after "1981 to 2017".                            |

| 144 import it as soon as possible.                                                                                                                                                                                                                                                   | eleased today, we will                                                                      |
|--------------------------------------------------------------------------------------------------------------------------------------------------------------------------------------------------------------------------------------------------------------------------------------|---------------------------------------------------------------------------------------------|
|                                                                                                                                                                                                                                                                                      |                                                                                             |
| 145 https://www.neotomadb.org/news/15/902                                                                                                                                                                                                                                            |                                                                                             |
| 146 Link to Neotoma Explorer added.                                                                                                                                                                                                                                                  |                                                                                             |
| 147                                                                                                                                                                                                                                                                                  |                                                                                             |
| -> A regression model for tree PAR vs forest cover is presented.                                                                                                                                                                                                                     | The regression is based on                                                                  |
| selected tree PAR values for 3% wide forest-cover bins (Figure 3)                                                                                                                                                                                                                    | b). The regression model                                                                    |
| suggests that an 80% forest cover within 10 km radius results in t                                                                                                                                                                                                                   | tree PAR values $> 3200$                                                                    |
| 151 (Conclusions, P23 L 15). I might miss an important point, but it s                                                                                                                                                                                                               | seems to me that the                                                                        |
| deduction is not supported by the data presented in Figure 3b. The                                                                                                                                                                                                                   | e Figure shows that values                                                                  |
| greater than 3200 tree PAR can be found even for 0% forest cove                                                                                                                                                                                                                      | er. It seems to me, instead,                                                                |
| that tree PAR are $> 20,000$ for forest cover $> 20\%$ (though striking                                                                                                                                                                                                              | gly the two sites with highest                                                              |
| 155 forest cover show rather low tree PAR values).                                                                                                                                                                                                                                   |                                                                                             |
| 156                                                                                                                                                                                                                                                                                  |                                                                                             |
|  <li>We believe that there is value in looking at the minimum values a</li> <li>well possible to obtain much higher PAR values at 80% forest cor</li> <li>3200 grains per cm2 are unlikely.</li>                                                                            | as clarified above. So yes it is
ver, but values lower than                              |
| 100 This relationship is immentant for intermediations on us often i                                                                                                                                                                                                                 |                                                                                             |
| 160 This relationship is important for interpretations as we are orien in
161 conservative interpretations on this "minimum" side of the spectr                                                                                                                                   | rum a g. during the initial                                                                 |
| 161 conservative interpretations on this minimum side of the spectr                                                                                                                                                                                                                  | un e.g. during the initial                                                                  |
| 162 spread of forest after the fee age.                                                                                                                                                                                                                                              |                                                                                             |
| $164 \rightarrow $ the manuscript shows decreasing pollen deposition as a function                                                                                                                                                                                                   | on of increasing distance to                                                                |
| 165 the nearest distribution limit of the parent plant species (Figure 4)                                                                                                                                                                                                            | ) Based on this evidence a                                                                  |
| 166 long-distance transport threshold (LDT) for a distance of 200 km                                                                                                                                                                                                                 | beyond the distribution limit                                                               |
| 167 is calculated. The thresholds could be used to infer range-size cha                                                                                                                                                                                                              | anges based on fossil PAR                                                                   |
| 168 values, the manuscript reports.                                                                                                                                                                                                                                                  |                                                                                             |
| - While these are interesting results and a potentially useful appro-                                                                                                                                                                                                                | bach, some critical discussion                                                              |
| 170 of this may be useful. The distribution limits were extracted from                                                                                                                                                                                                               | GIS shapefiles published by                                                                 |
| 171 Caudullo et al. (2017), which are publicly available on the figshar                                                                                                                                                                                                              | re website with associated                                                                  |
| 172 DataCite link (Caudullo, Giovanni; Welk, Erik; San-Miguel-Ayar                                                                                                                                                                                                                   | nz, Jesús (2017):                                                                           |
| 173 Chorological maps and data for the main European woody specie                                                                                                                                                                                                                    | es. figshare. Collection.                                                                   |
| 174 https://doi.org/10.6084/m9.figshare.c.2918528.v5).                                                                                                                                                                                                                               |                                                                                             |
| 175 We cite the version we used - $v2$ .                                                                                                                                                                                                                                             |                                                                                             |
| 176                                                                                                                                                                                                                                                                                  |                                                                                             |
| 177 In the original manuscript where Caudullo et al. present the maps                                                                                                                                                                                                                | s, they specifically mention                                                                |
| that "Since the maps aim at representing the species general chord                                                                                                                                                                                                                   | ology at continental scale,                                                                 |
| 179 providing a synthetic overview of distribution range, the mapped                                                                                                                                                                                                                 | boundaries should not be                                                                    |
| 180 considered as precise and sharp limits where the species is definit                                                                                                                                                                                                              | tely present or absent,                                                                     |
| 181 particularly at local level. Indeed, the first version of this dataset                                                                                                                                                                                                           | was created for the European                                                                |
| 182 Atlas of Forest Tree Species [16] to concisely outline the distribut                                                                                                                                                                                                             | ition ranges of described                                                                   |
| species, complementing information on the species biology and e                                                                                                                                                                                                                      | cology. Errors and                                                                          |
| 184 imprecision are partly inevitable, due to various causes, such as th                                                                                                                                                                                                             | he quality of the original                                                                  |
| source, the geo-referencing procedure, the interpretation and the c                                                                                                                                                                                                                  | comparison of the sources in                                                                |
| use same area and many due to the inmited precision of the manif                                                                                                                                                                                                                     | an digitalization process of                                                                |
| 107 the range harders (Fig. 1) "                                                                                                                                                                                                                                                     |                                                                                             |
|  <li>the range borders (Fig. 1)."</li> <li>We are aware that these distribution mens have uncertainties. Here</li>                                                                                                                                                          | wayor DADs are unlikely to                                                                  |
|  <li>the range borders (Fig. 1)."</li> <li>We are aware that these distribution maps have uncertainties. How</li> <li>be much effected by the occurrence of parent trees at yory low of</li>                                                                                | wever, PARs are unlikely to                                                                 |
|  <li>the range borders (Fig. 1)."</li> <li>We are aware that these distribution maps have uncertainties. How</li> <li>be much effected by the occurrence of parent trees at very low ab</li> <li>we did not look at the western distribution limits, but only explore</li>  | wever, PARs are unlikely to
oundance. Due to plantations
e latitudinal limits of tree |

192 193

measured by Abraham et al. truly represent the actual distances to the species distribution 194 limits. Thus, the precision and accuracy of the LDT values may be strongly overestimated and 195 misleading. Instead of using one single distribution limit, a range of distribution limits may 196 197 better represent the uncertainty of the mapped limits. Question is therefore: what LDT values would be obtained if the distribution limits of Caudullo et al. (2017) had an uncertainty? Say 198 ca. +100km or even +200km? 199 Since we focus in these comparisons on the northern distribution limits we believe that rare 200 occurrences of trees will be less and less likely as further north we go. So a given uncertainty 201 in the maps would not change the relative differences between taxa nor the order of 202 magnitude of the absolute values, which in any case are gross estimates. Still these numbers 203 are useful as modern analogues and represent guidelines rather than hard thresholds. 204 205 206 - Moreover, it would be useful to show the complete data in the plots of figure 4, including 207 the PAR values within the distribution range (thus extend the x-axes of the plots to include negative x-axis values). In theory at least, these PAR values should be greater than PAR 208 values around the distribution limit and beyond the limits. 209 We included the data for all traps in Fig. 4a). Within the area of distribution area the data is 210 represented as boxplots outside as a dots. 211 212 213 -> The comparison between modern and fossil PARs is interesting (paragraphs 3.3 and 3.4). - Paragraph 3.3. should be deeply revised and could be shortened. It could focus more on 214 PAR-inferred presence/absence based on LDT limits that were presented previously (Figure 215 5), and on the identification of the closest modern counterparts of populations sizes and forest 216 cover. Currently, some statements are descriptive and their relevance could be made clearer 217 (for instance, on P13 L31 "Modern and fossil values agree for the sites in central Sweden at 218 PARs between 1900-5600 grains..."). Some phrases could be removed (e.g. P13 L27 "As 219 discussed in the main manuscript, "), other ones are unclear (e.g. P13 L25 "ignoring traps 220 from the Caucasus and Turkey"), and several statements should be supported with references 221 to the literature (e.g. P16 L12"Picea abies is planted in many European regions outside its 222 natural distribution", or" Fagus pollen occurs at fossil sites that were assumed to have never 223 been within the distribution of the tree"), to name few examples. 224 Paragraph 3.3 was introduction for 3:4. It was shortened and included in the corresponding 225 226 section. The summarizing sentences we removed. 227 - Paragraph 3.4 is enlightening. However, the term "analogue" (and "modern analogue", 228 which is used later in the Discussion) is not appropriate. With pollen records, modern 229 analogues are generally referred to pollen assemblages (thus to vegetation composition). Here 230 instead you refer to "comparable, similar, population size of one taxon". Using the term 231 "modern analogue" without clarifying that you are using it with a different meaning creates 232 confusion, particularly in the Discussion where reference is made to the early-Holocene hazel 233 234 maximum. 235 We agree that the term "modern analogue" evokes comparison of assemblages of pollen percentages (sensu Overpeck et al., 1985), however we, similarly to Overpeck et al. (1985), 236 think that "modern analogue" should not be reduced to comparisons of modern pollen 237 assemblages, but describes any link between a modern situation and its resemblance of a 238 fossil find, including PAR. Also in geology and macro-ecology the term "modern analogue" 239 is used in a broader context as exemplified by these publications: Sidder, A. (2020), Ancient 240 sea levels in South Africa may offer modern analogues, Eos, 101, 241

- It is therefore highly questionable as to whether the distances to the distribution limits

| 242        | https://doi.org/10.1029/2020EO147001.; Horsák, M., Chytrý, M., Hájková, P., Hájek, M.,           |
|------------|--------------------------------------------------------------------------------------------------|
| 243        | Danihelka, J., Horsáková, V., Ermakov, N., German, D. A., Kočí, M., Lustyk, P., Nekola, J.       |
| 244        | C., Preislerová, Z. and Valachovič, M.: European glacial relict snails and plants:               |
| 245        | environmental context of their modern refugial occurrence in southern Siberia, Boreas, 44(4),    |
| 246        | 638–657, doi: 10.1111/bor.12133 , 2015.                                                   |
| 247        | We did look for a different terminology but could not find a better term for what we are         |
| 248        | comparing and clarified this in the introduction.                                                |
| 249        |                                                                                                  |
| 250        | -> There are other proxies that may be useful to determine presence of trees in fossil records   |
| 251        | (plant macrofossils and stomata). This could be mentioned in the text. Moreover, using fossil    |
| 252        | sites where such data is available could be useful to actually test the LDT limits, at least for |
| 253        | some of the taxa. Further, how do the inferences based on the LDT limits compare with            |
| 254        | inferences made previously based on pollen percentages (or on plant macrofossils and             |
| 255        | stomata)? For instance, in a prior study (Giesecke et al., 2017 in JBiogeogr) a good agreement   |
| 256        | between the estimates of overall spread (Fig. 5a) based on different pollen percentage           |
| 257        | abundance classes was found. Some of the fossil sites were actually analysed for pollen,         |
| 258        | stomata, and plant macrofossils (e.g. Sägistalsee, Bachalpsee), but these results are not        |
| 259        | mentioned in this manuscript.                                                                    |
| 200        |                                                                                                  |
| 261        | We compare LD1 limits with dispersal model as required be second reviewer. Other proxies         |
| 262        | is an interesting aspect, which indeed we did not address in this manuscript. In making the      |
| 263        | database accessible we nope that questions like these will lead to additional usages of the      |
| 204
265 | and text and to the current manuscript                                                           |
| 205        | and text and to the current manuscript.                                                          |
| 267        |                                                                                                  |
| 268        |                                                                                                  |
| 200        | Detailed comments:                                                                               |
| 205        | - title: influx or PAR? In the text you use PAR not influx Please homogenise terminology         |
| 271        | We use PAR in all text.                                                                          |
| 272        |                                                                                                  |
| 273        | - Please consistently use italics for latin names.                                               |
| 274        | I hope that all latin names are ok now, however translation via MarkDown is little bit tricky.   |
| 275        | P6 I 21 22: you artracted forest cover data from all grid calls within 10km radius of the        |
| 270
277 | trans. Did you then calculate a mean forest cover value? Please clarify                          |
| 277
278 | Ves clarified                                                                                    |
| 270        |                                                                                                  |
| 280        | - P7 L8: why 271 modern samples? The abstract mentions 2742 annual samples                       |
| 281        | While the database contains all data that was submitted, we only considered average trap         |
| 282        | record with at least 3 years and thus obtained 271 traps that we base the analysis on            |
| 283        | Averaging of the annual samples mentioned in the Methods. Number of traps and trap values        |
| 284        | is mentioned in the Results.                                                                     |
| 282        | - P7 I 10-ff · am having trouble understanding what has been done, and why. Did you match        |
| 286        | fossil and modern PAR based on their ranking in the classes? What has been averaged? why         |
| 287        | should 500-year bins represent periods of long-term vegetation stability (fossil pollen records  |
| 288        | show fast population doubling times for some taxa)? What distance measure did you use for        |
|            |                                                                                                  |

the cluster analysis (Euclidean distance)?

| 290
291                      | Rewritten. "periods of long-term vegetation stability" deleted, "distance matrix" changed to "matrix of distances".                                                                                                                                                                                                                                                                         |
|---------------------------------|---------------------------------------------------------------------------------------------------------------------------------------------------------------------------------------------------------------------------------------------------------------------------------------------------------------------------------------------------------------------------------------------|
| 292
293
294               | - P10 L111-12: total PAR is not shown in Figure 5.
Changed to "tree PAR"                                                                                                                                                                                                                                                                                                                 |
| 295
296
297
298        | - P11 L12: here the conclusion is presented before the data and then the data are presented to support the initially stated conclusion. Please reverse the line of arguments. Removed                                                                                                                                                                                                       |
| 299
300
301
302
303 |  <li>P11 L13: am having trouble to understand why 92 pairs were obtained. please clarify.</li> <li>We have 15 taxa in 7 regions, which make 105 cases that we investigate, however some regions lack species in the fossil and/or trap record and thus it is only possible to perform the t-test for 92 pairs of trap and fossil sites. We hope with Table S5 it is clearer.</li>  |
| 304
305
306               | - P11 L14: cannot find t-test and p-values on figure 6. Neither were "t-test" and "p-values" mentioned in the Material and Methods section. Please clarify why and what has been done.                                                                                                                                                                                                      |
| 307
308                      | We are sorry for this omission and will add it to the Methods section and add a table with p-
values to the Supplementary Table S5.                                                                                                                                                                                                                                                      |
| 309
310
311               | - P20 L12: MAT has a very low influence on pollen deposition (see Table S4). "strong influence" changed to "some influence".                                                                                                                                                                                                                                                                |
| 312
313
314
315        | - P20 L17-18: Which data/analysis/result shows that biomass cannot explain the latitudinal PAR gradient? changed to "forest cover"                                                                                                                                                                                                                                                          |
| 316
317
318
319        | - P 20 L31: why cannot the details be discussed here?
We meant that we can not discuss details of all species in all regions, our aims are general patterns. Sentence removed                                                                                                                                                                                                            |
| 320
321
322
323        | - P21 L14: statement "modern PAR for Betula and Pinus are not found in fossil samples" contradicts a prior statement (P10 L12 "Maximum PAR in traps are always higher compared to fossil situations, with the exception of Corvlus")                                                                                                                                                        |
| 323
324
325
326        | P10 L12 refers to the whole dataset, i.e. maximum at continental scale, P21 L14 refers to the situation of individual traps. In both cases comparison results in higher trap value than fossil one. We added exact values to make it clear.                                                                                                                                                 |
| 327
328
329
330        | - P22 L16: the effects are possibly small in relation to the wide environmental gradient.
We added: Also the comparison of values over this large environmental gradient results in the signal being stringer than the noise                                                                                                                                                             |
| 331
332
333
334        | - P22 L34-35: these are interesting notions. Please mention the usefulness of modern PARs (as listed here), and their importance for ecological and biogeographical studies, in the Introduction.                                                                                                                                                                                           |
| 335
336
337
338
339 | We added in Introduction: More recent investigations demonstrate the linear response of the absolute pollen deposition to absolute tree abundance (Seppä et al. 2009, Sugita et al. 2010, Matthias and Giesecke 2014), which may be used to reconstruct past standing tree biomass of different trees.                                                                                      |

340 - P22 last line: remove "and it is almost surprising that...explanatory power". It might be surprising, but is a fact that seems to contradict a prior study (Matthias et al.). Could the 341 gradient length be an important factor here? 342 Sentence removed. Yes, present study uses large gradient and all trees together, whereas 343 Matthias et al analyzed pollen-vegetation dataset by individual species. 344 345 -> Figures 6-13: - the a) and b) frames could be merged by using horizontal boxplots (instead 346 of barplots) in a), and adding b) as an overlay; 347 I have tried to use horizontal boxplots (Fig. R1), but the highest values enlarge the x-axis 348 while the lowest classes are not visible. The way of our presentation allows to appreciate the 349 general pattern of mean values for trap areas (a) and we make the full variability of individual 350 traps visible within the trap areas by coloured squares (b). 351 - font sizes are too small; 352 Font was enlarged to 2.5 for names of the fossil sites/trap areas and 2 for rest of the text. 353 354 355 -> References: - add: Caudullo, G., Welk, E., San-Miguel-Ayanz, J., 2017. Choro-logical maps for the main 356 European woody species. Data in Brief 12, 662–666.https://doi.org/10.1016/j.dib.2017.05.007 357 358 added -> In Table S4: 359 - caption: what does "alternatively" mean here? Please clarify. 360 "alternatively" removed 361 362 - Forest biomass, or forest cover (as of P6 L20-21)? 363 Changed to forest cover. 364 - are the "Adjusted R2" values of the "adjusted PAR" values the ones obtained with the 365 Andersen correction factors (P6 L23-24)? Please clarify. 366 Changed to: "(or logarithm of total PAR adjusted by Andersen values, Table S2)" 367 - please add r2 values for "tree PAR" (not total PAR) vs forest cover 368 369 Added 370 Anonymous Referee #2 371 Note to editor: The line numbers in the manuscript appear to be sometimes inconsistent. I 372 assume this is a fault with the submission process not the authors' fault. Apologies if any 373 confusion arises from this. 374 375 General comments An interesting manuscript with great potential to improve interpretations of fossil pollen 376 records. It is clear that a lot of careful thought and work has gone into this study. While it has 377 potential to be very useful, the structure and clarity of the work could be improved. I think in 378 particular the structure could be refined to be more consistent throughout, and in the 379 Introduction and Discussion sections, sub-headings introduced to clarify the development of 380 the argument. There seem to be various strands to this paper: 381 1. The correlations of PAR with parameters such as forest cover and temperature as calculated 382 from the modern samples 383 384 2. The question of long distance transportation beyond the current extent of the parent plant 385 taxa

| 386
387
388                                                                                                  | 3. The relationship between modern PAR and fossil PAR for selected taxa.                                                                                                                                                                                                                                                                                                                                                                                                                                                                                                                                                                                                                                                                     |
|--------------------------------------------------------------------------------------------------------------------|----------------------------------------------------------------------------------------------------------------------------------------------------------------------------------------------------------------------------------------------------------------------------------------------------------------------------------------------------------------------------------------------------------------------------------------------------------------------------------------------------------------------------------------------------------------------------------------------------------------------------------------------------------------------------------------------------------------------------------------------|
| 389
390
391
392                                                                                           | I feel that if the first two are clearly and separately addressed, it would be easier for the authors to address the third point coherently. At the moment, the results broadly follow this structure, but the Introduction and Discussion do not, I suspect if they did, the paper would flow better                                                                                                                                                                                                                                                                                                                                                                                                                                        |
| 393                                                                                                                | We are grateful for this suggestion and we re-structured the Introduction and Discussion.                                                                                                                                                                                                                                                                                                                                                                                                                                                                                                                                                                                                                                                    |
| 394                                                                                                                | To the State of this suggestion and the re-state are included on and Discussion                                                                                                                                                                                                                                                                                                                                                                                                                                                                                                                                                                                                                                                              |
| 395                                                                                                                | Specific points by section                                                                                                                                                                                                                                                                                                                                                                                                                                                                                                                                                                                                                                                                                                                   |
| 396                                                                                                                | Introduction                                                                                                                                                                                                                                                                                                                                                                                                                                                                                                                                                                                                                                                                                                                                 |
|  <li>397</li> <li>398</li> <li>399</li> <li>400</li> <li>401</li> <li>402</li> <li>403</li> <li>404</li>  | I was a little surprised that so much of the introduction dealt with history. Although I think
this would merit its own sub-section, I think it would be better to focus on scientific questions
addressed in the manuscript. Why pollen traps are an appropriate analogue for fossil records
should be introduced. It would be nice to see some mention of species ranges and their
possible fluctuation over time, and whether present day species distributions can be
considered to be in equilibrium with climate. Another factor that could be addressed in the
introduction is pollen dispersal and deposition; how far does pollen usually travel? This would
set up the argument for your chosen LDT threshold |
| 405                                                                                                                | set up the argument for your chosen LDT threshold.                                                                                                                                                                                                                                                                                                                                                                                                                                                                                                                                                                                                                                                                                           |
| 406
407
408                                                                                                  | We appreciate this suggestion to improve the organisation of the manuscript and we moved
the historical aspects of the development of the network (see earlier response) to a subsection.                                                                                                                                                                                                                                                                                                                                                                                                                                                                                                                                                 |
| 409

417                                                        | 2 Methods
General comment: all botanical names including species epithets need to be written
with their authorities the first time they are mentioned in the manuscript. Up to date
authorities can be found here http://www.worldfloraonline.org/. Upon first use, a species must
be written out in full even if its genus has been mentioned by name previously. This is to
avoid confusion between genera that start with the same letter. So for in-stance, Pinus
sylvestris L. could be shortened to P. sylvestris, but then Pinus mugo Turra needs to be
written as such before it can be abbreviated to P. mugo                                                                                                  |
| 418
419
420
421
422                                                                                    | We do not deal any extinct organisms in the past, but palaeoecology of living, abundant and well-known plant taxa. Biogeosciences is not taxonomical journal and many papers in Biogeosciences are published without authorities. Journal's guidelines do not mention the need for taxonomic authorities, but of course we can add them if our assoc. editor requires them.                                                                                                                                                                                                                                                                                                                                                                  |
| 423                                                                                                                |                                                                                                                                                                                                                                                                                                                                                                                                                                                                                                                                                                                                                                                                                                                                              |
| 424
425
426
427                                                                                           | Figure 1: As there is so much overlap on the map between modern and fossil sites, I think separating this out into two side by side maps with one showing fossil and one modern samples would make it clearer, and would also make it easier to go back and check locations of fossil sites as I was trying to do so later in the manuscript.                                                                                                                                                                                                                                                                                                                                                                                                |
| 428
429                                                                                                         | Thank you for this comment. We already separated a map of only fossil sites and it is better readable.                                                                                                                                                                                                                                                                                                                                                                                                                                                                                                                                                                                                                                       |
| 430
431                                                                                                         | On page 7, 2742 samples were mentioned as being in the database (Section 3.1) button line 8 only 271 are mentioned- which number was in the analysis?                                                                                                                                                                                                                                                                                                                                                                                                                                                                                                                                                                                        |

- We added to Section 3.1: "Considering the trap record with 3 years and more we obtained 271
  mean trap assemblages."
- 434 2.2 Data Collection
- 435 It would be interesting to see a plot of surface area of trap against PAR to test for a
- 436 relationship there and potentially be able to correct if one exists.

437 This is an interesting point and the reviewer is invited to try this out as all this information

- 438 will be available. Do you have any idea which mechanism can stay beyond?
- 439 2.3 Investigated taxa and Environmental parameter
- 440 Why was 200km chosen as the threshold for LDT?
- 441 Regional pollen is assumed to correspond to the vegetation cover in 100 by 100 km
- (Hellmann et al 2008) so doubling that distance represents a good rational. Additionally we
- 443 considered the uncertainty of the maps.
- 444
- 445 2.4 Comparison
- 446 Page 12: Figure 5 is a bit tricky to interpret, however, once I had realised what it was
- supposed to be showing I saw its value. I particularly like the LDT cut-off, which will be
- 448 potentially very useful in interpreting fossil records. I was surprised, however, that LDTs did
- 449 not receive further attention in the discussion section as it seems that they are a tangible,
- 450 useful output from the work.
- We added example of LDT cut-off on the end of the section Analogues for vegetation
   reconstruction.
- 453 3.2 Recent and fossil PAR
- 454 Figures 6 onwards: In caption, specify that distribution of taxon is in grey.
- 455 Added: "... distribution of the species (gray, Caudullo et al., 2018 ..."
- 456 These taxon-specific figures are in general, I think, quite useful. I hope they are reproduced a
- 457 little bigger (at least a page each)
- 458 Unfortunately one page per figure would surpass our APC budget.
- so that the details on the maps are easily readable. If this is not possible, maps should perhapsbe split into separate figures to improve legibility.
- We will improve legibility by abbreviating names of fossil sites, so we make use of blankspace and the map is larger now. We will enlarge all text in the figure to 2 and 2.5.
- 463 I am not sure the multi-coloured coding for the PAR values adds much to the figure-
- 464 Multi-coloured coding helps to link the fossil and trap PAR values of the same height/class. In
- the fossil stratigraphic plot we point out the class of interest, coloured squares in 6b) illustrate
- the variability of traps within one trap area.
- 467 you could probably do away with (b) and still retain the meaning of the figures.

- Boxplot would be one option as proposed by reviewer 1 (see response there), black and white 468 bar-plots for the fossil and trap record without any colours would not visualize the analysis we 469 did. 470 I am also not clear on why, in the fossil graphs, only certain colours are included 471 472 - why is only the highest PAR of interest? Only certain classes appeared in the fossil record. We picked the highest PAR class from the 473 474 fossil record, because it represents also the densest population of the source plants. 475 3.4 Taxa specific results How were the 'main taxa' to be presented chosen for this section? It 476 seems odd that some are arbitrarily in the SM, particularly arboreal pollen which was 477 478 presented in Figure 2. Figures need to be referred to consistently in this section. We state in Methods that they are/were dominant. In 3.3 Section we list them. We do not want 479 to flood reader by all species. Fossil-trap links of species selected for the main text show 480 nicely changes of distribution patterns, whereas species in the SM can suffer from certain 481 biases and limitations (See Cyperaceae and Poaceae in section 4.2) 482 483 3.4.2 Betula Why are traps from the Caucasus and Turkey ignored? 484 Reworded to "Letting aside traps from the Caucasus and Turkey...(, because those two areas 485 have different species Betula than rest of the trap areas, but the most we deal with belong to 486 B. pendula and B. pubescens)" 487 4.1 Discussion 488 The first sentence of the discussion doesn't seem to tally with the results- it looks, from your 489 data, like the relationship between modern and fossil PAR is actually quite complex and 490 variable. I don't necessarily think this is a bad thing, however; the paper presents a 491 quantitative dataset that could potentially be used to help researchers quantify what their 492 PARs from fossil data actually mean. 493 We included mention about complexity of the relationship to the first sentence: "in spite of 494 different taphonomic processes that influence PAR values in pollen traps versus lake 495 sediments.". 496 497 Line 22 onward: This paragraph seems to be about LDT, but that isn't explicitly stated. 498 Last paragraph of page 20 onwards appears to be, broadly, taxon-specific discussion of modern and fossil PARs; it would be better if this were clearly signposted and possibly split 499 up with sub-headings. 500 Discussion divided into a more subsections. LDT subsection added 501 I would be interested to see some consideration of how these results might be useful in 502 503 feeding into quantitative reconstructions of vegetation. Although PDD models tend to deal in
- 504 percentages, surely this approach (on cores with appropriate chronologies) could open the

- door to future models being calibrated using PARs, an interesting prospect for vegetationreconstructions, particularly given your LDT estimates.
- We included in the discussion possibilities of PAR vs. PDD and wider use of LTD estimates
  in quantitative reconstruction. See also Table S6.
- 509 4.2 Limitations
- Line 21: Why are only 3 fossil sites listed here? Are the others not likely to show any bias?
- 511 The fossil PAR values at the 3 sites are rather high so we suspect that lake internal processes 512 may explain these values. In generall the reviwer is right these biases may also occure at other
- 513 sites and we added "especially".
- 515 Line 28: Unclear which analysis only included Poaceae and Cyperaceae
- 516 Changed to "...Poaceae and Cyperaceae are only herbs selected for our analyses."
- 517 5 Conclusions
- 518 This is succinct and mostly well-structured but would benefit from a closing statement
- 519 outlining the applications and take home message of the paper.
- 520 We removed little bit vague last paragraph Conclusions and thus we make both ouputs, which
- are useful for palaeoecological applications PMP database and LDT trhesholds more
- 522 visible.

514

- 523 Technical comments
- 524 Page 2
- 525 Line 12: replace 'case' with 'cause'
- 526 Replaced
- 527 Line 13: Could be rephrased as 'hopefully serves to improve interpretations' (or remove
- 528 hopefully- I think it definitely will).
- 529 "hopefully" removed.
- 530 Page 3
- Line 1: A good recent reference here would be Haselhorst 2020 (DOI:10.1111/jvs.12897)
- showing high interannual variability in the tropics too- strengthens general argument.
- 533 Citation added.
- Line 7: remove comma after 'Although'.
- 535 Comma removed.
- 536 Page 7
- 537 Line 13: something strange happening here after 'values variation'- typo?
- 538 "variation.ignoring" replaced by "and we need to ignore"

- 539 Line 19: The sentences about fossil pollen seem out of place here as this section of the results
- 540 is regarding modern pollen.
- 541 The sentences about fossil pollen placed to section of the fossil pollen.
- 542 Page 14
- 543 Line 5: 'main text' where in the manuscript is being referred to?
- 544 "As discussed in the main text," removed
- 545 Page 20
- 546 Line 31 paragraph: I think this paragraph might be better placed at the end of this section.
- 547 Paragraph placed to the end of the section.
- 548 The phrasing in Line 32 seems a little odd as you go on to give an example of linking fossil to
- 549 modern PARs- perhaps delete the 'details cannot be discussed'sentence.
- 550 Rephased and "Unfortunately, the details cannot be discussed here." removed.
- 551 3.4.8 Line ? (Line numbers unclear) correct LTD to LDT and remove 'threshold'
- 552 Corrected, removed.
- 553
- 554

**Patterns in recent and Holocene pollen influxes Pollen Accumulation Rates across Europe; the Pollen Monitoring Programme Database as a tool for vegetation reconstruction**

Voitěch Abraham1, Sheila Hicks2, Helena Svobodová-Svitavská1, 3, Elissaveta Bozilova4, Sampson Panajiotidis5, Mariana Filipova-Marinova6, Christin Eldegard Jensen7, Spassimir Tonkov4. Irena Agnieszka Pidek8, Joanna Świeta-Musznicka9, Marcelina Zimny9, Eliso Kvavadze10, Anna Filbrandt-Czaja11, Martina Hättestrand12, Nurgül Karlıoğlu Kılıç13, Jana Kosenko14, Maria Nosova15, Elena Severova14, Olga Volkova14, Margrét Hallsdóttir16, Laimdota Kalnina17, Agnieszka M. Noryśkiewicz18, Bożena Noryśkiewicz19, Heather Pardoe20, Areti Christodoulou21, Tiiu Koff22, Sonia L. Fontana23, 24, Teija Alenius25, Elisabeth Isaksson26, Heikki Seppä27, Siim Veski28, Anna Pedziszewska9, Martin Weiser1, and Thomas Giesecke29 1Department of Botany, Faculty of Science, Charles University; Benátská 2; CZ-128 01; Prague; Czech Republic 2P.O. Box 8000, FI-90014 University of Oulu, Finland 3Institute of Botany v.v.i.; Czech Academy of Sciences; Zámek 1; CZ-252 43 Průhonice; Czech Republic 4Laboratory of Palynology, Department of Botany, Faculty of Biology, Sofia University, 8 Dragan Tsankov blvd., Sofia 1164, Bulgaria 5Lab. of Forest Botany, Faculty of Forestry and Natural Environment, Aristotle University of Thessaloniki, P.O. Box 270, 54124 Thessaloniki, Greece 6Museum of Natural History Varna, 41 Maria Louisa Blvd. 9000 Varna; Bulgaria 7University of Stavanger, Museum of Archaeology, Peder Klows gate 31A, PB 8600 Forus, NO-4036 Stavanger, Norway 8Institute of Earth and Environmental Sciences, Maria Curie-Sklodowska University: al. Krasnicka 2d; 20-718 Lublin; Poland 9University of Gdańsk, Faculty of Biology, Department of Plant Ecology, Laboratory of Palaeoecology and Archaeobotany, ul. Wita Stwosza 59, 80-308 Gdańsk, Poland 10Georgian National Museum, Purtseladze Str.3, Tbilisi 5, Georgia 0105. 11Faculty of Biological and Veterinary Sciences, Geobotany and Landscape Planning, Nicolaus Copernicus University in

Toruń, 87-100 Toruń, ul. Lwowska 1; Poland

12Department of Physical Geography, Stockholm University, SE-106 91 Stockholm, Sweden

13Department of Forest Botany, Faculty of Forestry, Istanbul University-Cerrahpaşa; Bahçeköy; TR-34473; Istanbul; Turkey

14Depertament od Higher Plants, Moscow State University; Leninskie Gory, 1, 12, Moscow, 119234, Russia

15Main Botanical Garden RAS; Botanicheskaya, 4, Moscow, 127276, Russia

16Laugarnesvegi 87 íbúð 105, 105 Reykjavík, Iceland

17Department of Geography and Earth Sciences, University of Latvia; Jelgavas Street 1, LV-1004; Riga, Latvia

18Institute of Archeology, Faculty of History, Nicolaus Copernicus University in Toruń; Szosa Bydgoska 44/48; 87-100 Toruń; Poland

19Faculty of Earth Sciences and Spacial Managment, Nicolaus Copernicus University in Toruń; Lwowska 1, 87-100 Toruń; Poland

20Department of Natural Sciences, National Museum Wales, Cathays Park, Cardiff, CF10 3NP, U.K.

21Department of Forests, Ministry of Agriculture, Rural Development and Environment, P. Box 24136, 1701 Nicosia, Cyprus

22Tallinn University, School of Natural Sciences and Health, Institute of Ecology, senior researcher. Uus Sadama 5, 10120 Tallinn, Estonia

23Cátedra de Palinilogía, Facultad de Ciencias Naturales y Museo, UNLP, Calle 64 n°3, 1900 La Plata, Argentina

24Faculty of Resource Management, HAWK University of Applied Sciences and Arts, Büsgenweg 1a, 37077 Göttingen, Germany

25Turku Institute for Advanced Studies (Department of Archaeology), FI-20014 University of Turku

26Norwegian Polar Institute, Fram Centre, N-9296 Tromsø, Norway

27Department of Geosciences and Geography, University of Helsinki, Gustav Hällströmin katu 2, 00014, Helsinki, Finland
 28Department of Geology, Tallinn University of Technology, TalTech, Ehitajate tee 5, 19086 Tallinn, Estonia

29Palaeoecology, Department of Physical Geography, Faculty of Geosciences, Utrecht University, P.O. Box 80115, 3508 TC Utrecht, The Netherlands.

Correspondence: Vojtěch Abraham (vojtech.abraham@gmail.com)

**Abstract.** The collection of modern spatially extensive pollen data are is important for the interpretation of fossil pollen diagrams. Such datasets are readily available for percentage data but lacking for pollen accumulation rates (PAR). Filling this gap has been the motivation of the pollen monitoring network, whose contributors monitored pollen deposition in modified Tauber-traps for several years or decades across European latitudes. Here we present this monitoring dataset consisting of 351

- 5 trap locations with a total of 2742 annual samples covering the period from 1981 to 2017. This dataset shows that elimate parameters correlating with latitude total PARs are influenced by forest cover and climate parameters, which determine pollen productivity . A signal of regional forest cover can be detected in the data, while local tree cover seems more important, and correlate with latitude. Pollen traps situated beyond 200 km of the distribution of the parent tree a given tree species are still collecting occasional pollen grains of the tree in question that species. PAR's of up to 30 grains cm-2 y-1 in fossil diagram
- 10 should therefore be interpreted as long distance transport . Comparisons to fossil data from beyond 200 km from the area of distribution. Comparisons between modern and fossil PAR from the same areas regions show comparable values. Comparisons often demonstrate that similar high values for temperate taxa in fossils sites are found further south or downhill. While modern situations comparable to high PAR values of some taxa (e.g. *Corylus*) may be hard to finddo not occur, CO2 fertilization and land use may ease cause high modern PAR's that are not documented in the fossil record. The modern data is now publically
- 15 available in the Neotoma Paleoecology Database and hopefully (https://apps.neotomadb.org/explorer/) and serves improving interpretations of fossil PAR data.

*Copyright statement.* The article is distributed under the Creative Commons Attribution 4.0 License. Unless otherwise stated, associated published material is distributed under the same licence.

[revised manuscript text omitted]

---

## Referee Report (RR1)

[referee-annotated manuscript omitted]

---

## Author Response (AR2)

Submitted on 29 Jan 2021
Referee #3: Simon Connor, simon.connor@anu.edu.au

**Anonymous during peer-review:** Yes No

**Anonymous in acknowledgements of published article:** Yes No

**Recommendation to the editor**

**1) Scientific significance**
Does the manuscript represent a substantial contribution to scientific progress within the scope of this journal (substantial new concepts, ideas, methods, or data)?

Excellent Good Fair Poor

**2) Scientific quality**
Are the scientific approach and applied methods valid? Are the results discussed in an appropriate and balanced way (consideration of related work, including appropriate references)?

Excellent Good Fair Poor

**3) Presentation quality**
Are the scientific results and conclusions presented in a clear, concise, and well structured way (number and quality of figures/tables, appropriate use of English language)?

Excellent Good Fair Poor

For final publication, the manuscript should be

**accepted as is**

accepted subject to **technical corrections**

accepted subject to **minor revisions**

reconsidered after **major revisions**

    I am willing to review the revised paper.

    I am **not** willing to review the revised paper.

**rejected**

**Suggestions for revision or reasons for rejection (will be published if the paper is accepted for final publication)**

This is a very interesting and challenging research topic that has the potential to open up new avenues for the accurate reconstruction of past vegetation and biomass patterns globally. By using an extensive dataset from standardised pollen trapping experiments around Europe, the manuscript attempts to provide thresholds for the local presence of a number of key tree species in the vegetation history of the region. By focussing on pollen accumulation rates, the authors provide the first continental-scale dataset that is relevant to reconstructing past changes in biomass, tree migration rates, palaeo-population dynamics and so on. The results could even be used to help anticipate the effect of climate change on pollen production in the future, which could have implications for allergy sufferers in the region, forest health and conservation. The manuscript includes some very involved and novel graphical presentations

in R and much relevant literature.

Unfortunately the manuscript is not yet ready for publication, in my opinion. The shortcomings are detailed below and are all quite easy to solve with some moderate revision of the text, particularly the Introduction and Methods, which currently lack coherent links to the study aims. I really hope the authors will make the effort to modify their manuscript, as this work is certainly novel and has great potential to be a turning point in palaeoecological and biogeographical research.

Major comments:

There is a surprising number of grammatical mistakes given the authors involved. Some of these mistakes are identified in detailed comments, below, but thorough proof reading would be helpful before resubmission.
The manuscript was read by native speaker.

The Introduction would benefit from tighter structuring and should emphasise the broader context for the research (i.e. beyond palynology - see suggestions in the first paragraph above). Perhaps breaking it into subsections, each dealing with one of the aims, would make the text more focussed, e.g. Relationships between PARs, climate and vegetation cover; The importance of long-distance transported pollen; Applicability of PARs to palaeovegetation reconstruction. At the moment, the Introduction seems to be largely directed at Aim 1, whereas Aims 2 and 3 are not adequately introduced. This leaves the reader wondering why certain methods of analysis were chosen later in the paper. The current text could be shortened in places without loss of information. Taphonomic processes that affect lake sediment pollen should be introduced in the section relating to fossil PARs, not at the end of the Discussion (Section 4.4), as these are critical to interpretation.
The first paragraphs were rewritten to emphasize the importance of this study and the three questions were separated into paragraphs introducing them as suggested.

In the Methods, it would also be good to split sections 2.3 and 2.4 into 3 sections dealing with each of the 3 aims. -> Done
Section 2.4 mentions selecting fossil sites within each trapping region, but were any criteria used to make the selection? Were records filtered according to taxonomic resolution or the number of radiocarbon dates? As PARs are highly dependent on accurate chronologies, some mention of chronological control should be made in this section.
We added constraints for the selection of sites. However, taxonomic resolution is not a criterium here as we only investigated abundant taxa that are usually identified by all investigators.
Table 1 should include the number of dates contributing to each site's chronology. Section 2.4 describes a statistical clustering technique that is poorly explained. Please give some more details to explain how the method works and why it is the most appropriate option for addressing Aim 3.
We changed the text to emphasize the purpose for using a one-dimensional clustering metode here. The method is widely used, referenced and details of how it works can be easily found elsewhere.
 A number of new methods are introduced in the Results and Discussion sections (see below) – these should first appear in the Methods.
All methods are described in the method section, some are mentioned again in the result section to remind the reader how the values were derived.

In Results, the "3% wide bin" approach requires more explanation and justification. It seems to be an arbitrary solution to finding a trend in the data, rather than based on any objective criteria.

The 3% wide bin was found exploring the data; it was not an a priori condition. We indicated this in the result section.

Please try to avoid this perception by making it clearer how and why this approach was taken. It would be best to add this to the Methods, rather than introducing new data treatments (i.e. the binning approach and regression analysis) here in Results.

In the result section on LDT we changed "regression analysis" for "fitting a linear relationship". This is also and was present in the method section; we only repeat it here to explain the values. We agree in general that the methods should be clearly stated in their section. However, it is sometimes easier to add detailed information on the data treatment with the description of the results.

The authors claim that a threshold of 30 grains/cm2/yr indicates long-distance transport (LDT), but Fig. 4 shows that only Picea and Quercus had LDT components below the error bars for traps within the species' geographic ranges. This is important to note because researchers could apply the threshold and mistakenly reach the conclusion that it denotes the absence of the species (a key question in island biogeography and post-glacial tree migration). This makes Fig. 5 potentially misleading. Maybe a better approach is to consider different thresholds for each species – the PAR threshold of 30 grains/cm2/yr might be appropriate for Fagus, but not for other taxa (Fig. 4b). The authors should also explain why 200 km is an appropriate distance when much better results are obtained at greater distances.

The choice of 200 km is now motivated in the method section. A note on Fagus was added to the result section as well as a general note of warning. A note of warning was also added to the discussion.

Section 3.4 includes a t-test for differences between fossil and trap PARs, but it is unclear why this was done and which aim it addresses.

The motivation was added to the method section and the text in the result section was revised to

Section 3.5 includes detailed descriptions of 8 different taxa. These are very long and include too much interpretation for a Results section. Consider removing these or reducing the descriptions in the manuscript to a single sentence per taxon and put the longer interpretations (existing text) in the supplement. The ad hoc exclusion of Betula data from Turkey and Georgia seems difficult to justify (see specific comments) and requires some explanation in the text.

Ok, we moved this description to the supplementary, extracted the most important information per species into a one sentence and created new Section 3.5. We wanted to express in the first sentence of the Betula section, that other species than B. pendula and B. pubescens grow in Georgia and Turkey. Traps from both areas were included. Rephrased.

The Discussion seems to say that taphonomic processes can be disregarded in comparing trap and fossil pollen.

The sentence in the beginning of the discussion states that modern and fossil values are comparable "in spite of different taphonomic processes", meaning that pollen collection in

the trap is markedly different from the accumulation in traps. The reference to Lisitsyna et al. (2011) comparing pollen accumulation in traps to lake was added.

It is hard to see how this can be claimed without a full explanation of those processes. The authors link pollen production to primary productivity gradients (latitude), yet do not consider how these gradients might have changed during the Holocene due to millennial-scale climatic variations.

We only compare the modern pollen trap data to the latitudinal gradient, not the fossil. Where we search for analogous fossil values to the PAR in the traps we do interpret these links with a changing Holocene climate.

This makes the comparison between fossil and trap data quite complex and these complexities should be assessed in the Discussion.

We mention Corylus and Pinus each as an example of taxa which is more abundant in past one in the present due to climatic variations.

The section about long-distance transport could also consider how elevation might affect trap results – a trap placed in Fagus forest will presumably catch more pollen than a trap on a treeless mountaintop, even though both traps are within Fagus's geographic range.

Elevation has little influence on the deposition of long distance transported pollen when considering PAR. There is a large influence when looking at percentages, but the amount of pollen coming from 200 km or beyond landing on the mountain top is similar to what should arrive in the valley.

We added two sentences in the Discussion. Even though, the present dataset contains several altitudinal transects, linear model between altitude and PAR does not have much sense, since variability between them are clearly larger. Good topic for next analysis by linear mixed models.

The use of a Gaussian Plume dispersal model is mentioned for the first time in the Discussion, but should be introduced in Methods.

We introduced this in Methods, added sentences in Results and improved sentences in the Discussion.

Section 4.3 has lots of potential to explain the importance of linking modern and fossil PARs, but gets very detailed very quickly, making it difficult to see the overall picture. Please try to broaden the scope here and use the site-specific details to support your argument. Help the reader understand why it's important that no modern analogues for early-Holocene Corylus PARs exist in Europe, for example.

The text has been rephrased to explain the importance of the examples.

The limitations and problems section (4.4) is well thought through and contributes substantially to the paper's scientific value. However, it leaves the reader perplexed as to what are the strongest points of the analysis. Which results/outcomes of the paper can we regard as being the most robust?

All the results and insights are useful. The LDT limit is probably the most robust single number. For the rest, you indicated above that uncertainties should be stated. Well here they are.

Specific line-by-line comments:

Page 2, line 1: Consider putting a comma after "modern" and replacing "diagrams" with "assemblages and the reconstruction of past vegetation communities in space and time" to expand the scope of the paper from a purely palynological one.

done

2: Replace "Such" with "Modern" [to avoid confusion]

done

4: "European latitudes" sounds strange as the same latitudes are found in N America and E Asia. Consider "Europe" instead

done

7: Replace "are still collecting" with "still collect"

done

9: "Comparisons… show comparable values" sounds strange – consider "similar values"

done

10: Replace "fossils" with "fossil"; this sentence is hard to understand – are the fossil sites located further south and downhill compared to the trap sites, or vice versa? What is meant by "similar high values" in this context. Please rephrase more simply

Rephrased to "Comparisons for temperate taxa often demonstrate that similar trap values are found further south or downhill."

11: The sentence "While modern… do not occur" is unclear. Do you mean that, for some taxa, PARs in the past were much higher than those recorded in the traps?

rephrased

12: Replace "PAR's" with "PARs" and "publically" with "publicly"

replaced

13: Replace "serves improving" with "serves to improve" or simply "aids"

replaced

19: I suggest adding a statement before the opening sentence that highlights the relevance of pollen analysis. This would provide a broader context for the paper and might attract non-palynological readers!

Sentence added.

21: Comma after "tree-line"

done

22: "procuring" should be "producing"

Changed according to major comment.

Page 3, line 1: place commas around the phase "or… period of time"; also note that "is better" should be "are better" to agree with rates

done

3: It would be useful to point out what makes this paper so groundbreaking, as the sentence seems lost without such elaboration

"afforestation" replaced by "spread of trees"

8: Inconsistent use of "PAR" vs "PARs" (cf. line 6, this page); change "sediments" to "sediment"

Corrected to PAR.

14: Comma needed after citation

done

16: Ditto

done

18: Replace "numerable" with "numerous"

done

19: Replace "comparably" with "comparatively"

deleted

21: "were based" should be "was based" to agree with construction.

done

26: "of the previous, as well as the year of flowering" – it is unclear whether this means the previous topic (tree biomass) or the previous year. Consider "of the year of flowering and previous year"

done

27: The question posed here does not arise from the previous statements. You state that pollen deposition rates represent absolute tree abundance, but then say that interannual PAR variations are determined by weather, so it is unclear why climate (which is different to weather) or site conditions (whatever that means) would raise questions. Please rephrase this to help readers follow your arguments

Sentences changed to "However, the absolute pollen deposition must be averaged from several years,…"

28: The sentence "Comparing… suggest…" would make more sense with "A comparison of… suggesting"; this sentence might be better placed before the question above to provide context

This was changed and moved.

30: What is the basis for interpreting the PAR:weather relationship as reflecting primary productivity of the tree? Isn't flowering (and pollen production) specific to the phenology of each species and may have many different weather triggers according to each species? See Autio and Hicks 2003 https://doi.org/10.1080/00173130310017409. Your primary productivity theory cannot explain masting or trees that flower in response to stress. Please rephrase this sentence and the following one and include some references in support of your claims. What's the basis of the CO2 argument?

Whole paragraph rewritten, reference added.

32: Delete "Already" [awkward] or replace with "As early as the 1940s"

done

34: Place a comma either side of "however"

done

35: Do you mean "then" rather than "than"?

done

How is the climatic interpretation here different to what Davis and Deevey proposed?

Sentence changed: "could also be due to a change in these parameters and not only due to tree abundance"

Page 4, line 1: The "initial question" has not been introduced previously. Please elaborate on this and explain to the reader why it is important to determine the long-distance component more accurately, including references (e.g. Markgraf 1980, Grana 19, 127-146). It's a very important question for island palaeoecology, where the presence of absence of pollen can often be used to decide if a species if native or exotic (e.g. http://doi.org/10.1111/j.1365-2699.2008.02012.x, and http://doi.org/10.1126/science.1163454). Splitting the long-distance aspect off into a standalone paragraph might be a good idea.

The "initial question" was introduced by Hesselman 1919, cited here as a second study. New paragraph produced.

7: Delete "as references helping" [unnecessary]

done

8: Comma needed after "Finland"

done

9: Change "applied" to "applied to"

done

16: Add comma after citation

done
20: Ditto
done
32: This aim seems fine, but the preceding introduction adds a lot more variables, such as site condition, CO2 fertilization, weather of the flowering season etc. Perhaps you could explain why only climate and forest cover are retained in the aims?
Because they were not studied, the weather was. We hope this decision is clear from paragraph 1.2.
Page 5, line 3: The 3rd aim does not seem to arise so easily from the introduction. Could you perhaps provide some more context in the introduction to tell readers why this aim is important and necessary?
This is introduced by subsection "Modern analogies"
8: Replace "the" with "their" or simply delete; I suggest putting the names of the trap regions in this section in capital letters, e.g. "North Boreal" instead of "north boreal" to make them stand out more
done, capitalized in whole text
Page 6, line 21: Comma after "overview"
done
29: Replace "In consequence" with "Consequently,"; replace "might overgrow or cover" [future tense] with "might have overgrown or covered" [past tense]
done
Page 7, line 5: Provide a citation to Tauber's paper where these components are described.
Tauber 1967 provided
29: Add a comma after "PAR"
done
32: "in the pollen type described above" – perhaps "in each of the pollen types listed above" would be clearer
done
34: "these taxa" – do you mean the taxa not suitable for comparison, or the taxa that were suitable? Unclear. It is also unclear how the pollen traps were placed at exactly 200 km from the edge of the plant distribution limits. Do you mean >200 km? Or within 200 km?
Yes, this was unclear: suitable replaced with possible, and rephrased to: "Linear regression between this distance and the decadic logarithm of PAR was used to explore thresholds of long-distance transport (hereafter also as "LDT") at 200 km from their mapped distribution limits."
Page 8, line 2: Maybe "target taxa" instead of "taxa considered"
done
4: Why was log10 PAR used instead of PAR? Justify
We expect a logarithmic decline of PAR away from the source and added this observation: "Initial observations showed that PAR dropped rapidly away from the distribution of the parent tree and did not decline at the same rate at larger distances. We therefor compared distance to the decadic logarithm of PAR, applying linear regression to explore thresholds of long-distance transport (hereafter also as "LDT")."

8: Replace "Per" with "For"
done
10: Add comma after "comparison"

done

12: "logged PAR" – do you mean "log-transformed PAR"? It is unclear how this sentence compares traps and fossil data – it seems to only deal with trap data – please expand.

Expanded and changed substantially by your major comments. Log-transformed and log-normal in whole text.

15: "at level" – replace with "at the level".

done

The description of the methods here is wordy but does not really convey why one-dimensional clustering was the most appropriate method and what statistical criteria were used to form the clusters. More detail about the method would be useful.

Sentence "This method splits the univariate data in the way that the total of within-cluster sums of squares is always minimum." added

"The classes produced were used to facilitate the comparison between trap and fossil data and to match the trap values with analogous situations in the past. The aim of this comparison was to find traps with similarly high values for individual taxa that compared to the highest average fossil PAR" – these sentences are quite wordy and seem to be saying "These classes helped us compare trap and fossil data and to link high trap PARs with high fossil PARs of the past".

Changed.

Please explain why only high values were considered meaningful for comparison.

"We dealt only with the highest class in each fossil sequence, because maximum abundance of several our target taxa was used as a stratigraphic marker of the Holocene period."

19: The grammar of this sentence needs attention

done

23: supplementary material?

deleted

27: How does a "mean trap assemblage" differ from a "trap location"?

"Considering the trap record with 3 years" we excluded some traps. Here we inform about the number of traps in the database and then the number of traps in the analysis.

28-30: These sentences describing climate and elevation might be more appropriate for the Study Area section (2.1)

Yes and no, we understand it as a result of our data extraction from the climatic dataset. The range of elevations was obtained after the compilation of all traps in the database, thus we hope it can stay in results as well.

Page 9, line 2: Comma after "environments"

done

3: Comma after "type"

done

6: Explain briefly what makes these differences noticeable

Changed to: "Dominance of oak and hornbeam at Temperate/Mediterranean sites in the lowland and pine and birch at Arctic/Alpine and North Boreal sites show similar stability in the Holocene perspective. Vegetation history at the rest of the fossil sites show more dynamic development"

10: Delete "Nevertheless"; hyphenate "log-transformed";

done

the text here refers to "total PAR" but the figure referred to (Fig. 2) only displays "tree PAR" – please indicate where the total PAR data can be found

referred to tree PAR

12: How much variance did elevation explain on its own? This information is missing from Table S4

Relationship of the elevation and tree/total PAR in such a large dataset would need to be tested by mixed models, which would allow to separate different transects in Alps, Georgia, Pieria, Šumava, Krkonoše... Nice idea, but it would be too much for this paper. Simple linear model show that elevation itself explains less than 5%.,

Page 10, Fig. 2: This is a very comprehensive figure! However, the absence of data for the Georgian sites should be explained somewhere in the text. Vegetation data were collected for the Georgian pollen traps in Filipova-Marinova et al. 2010 (https://doi.org/10.1007/s00334-010-0257-z ).

Thank you, unluckily our data source does not cover Georgia and vegetation data do not contain general forest cover. Fig. 2 in Filipova-Marinova et al 2010 contains only some tree species.

Page 11, line 1: "affect" rather than "effect". It's unclear what relationship is meant here – clarify. Also, consider "tree species" to replace the potentially confusing wording "different trees".

done

4: Comma after "grasses". Why were grasses included here and not in the previous analysis? This is not mentioned in the Methods and should be explained somewhere.

We added to Methods: "...explain the variance in average pollen accumulation of total and tree PAR"

5: Hyphenate "log-transformed"

done

8: Replace "pattern, the" with "pattern, so the"

done

9: Replace "are" with "is" to agree with "distribution"; same problem in next line

done

15: What is meant by the "3% wide bins"? This seems to be a methodological decision that needs further explanation. How was the 3% value determined? Why not use a more 'usual' value like the lower quartile? What is the regression model referred to here? The sentence here is very difficult to understand and should be rewritten more clearly.

Changed to "Exploring the data showed that a 3 %.."

18: Is "PAR" singular or plural in this sentence? Perhaps using PAR for singular and PARs for plural would avoid confusion?

Singular grammar corrected

19: What is meant by "distribution limit" – the geographic distribution or distribution of points on the plot?

changed to: limit of geographic distribution

21: Comma after "taxa".

done-

Regression analysis should be mentioned in the Methods rather than Results.

It was at page 8 line 4. / we keep it there.

In this section, it would be useful to mention which taxa show a significant difference between LDT pollen and local pollen (within the species' range).

Yes, this comment is very important, thank you. It led us to reconsider the analysis for Fig 4. All regressions were significant, but we realized that we ignored values within the distribution area. Now, they are included. (All regressions are significant, of course). LDT is

higher and more correct.

26: Delete "here"

deleted

29-33: Where are these results presented?

added reference to Fig 5 again.

Page 12, line 1: Replace "Minimal" with "Minimum"

done

2: Commas after "Cyperaceae" and "particular". Please provide a supplementary figure that shows the distributions being described in this paragraph, or consider omitting this information about frequency distributions as its relevance is unclear.

Distributions are in Figure 5 in paired histograms.

3: It's hard to understand what is meant by "fossil PARs show a local maximum in the frequency of low values, which does not occur in the traps", especially since there is no associated figure.

Pointing out "local maxima" was deleted.

5: Commas after "types" and "these". It is unclear why this analysis was done or what the "pairs" are comprised of – please explain.

"pairs of trap and fossil data"

6: Comma after "comparison"

done

Page 13, line 3: Comma after "PAR"

done

5: What are "maximum averages"? Do you mean average maxima? Hyphenate "site-by-site"

Changed to highest class of fossil PAR

7: Why are results for all these 8 taxa included, while others are in supplementary material? How were the 8 taxa chosen and are they all important? Also replace "description" with "descriptions".

We followed your suggestion from major comments and we present all taxa in the main paper by one sentence. and in detail in the supplementary.

8: Missing word – "supplementary material" or use "supplement"

done

12: Delete "the" before "different"

done

15: Comma after "populations"

done

17: Comma after "Sumava"

done

Page 15, line 4: Explain why the Georgian and Turkish traps were excluded (this information is in the responses to reviewers, but not in the text). It's hard to understand why these countries were excluded because of the presence of other species of Betula, when the same approach was not taken to Pinus, Fraxinus, Fagus, Carpinus, Quercus – all of which have different species in the Caucasus and Anatolia. The map, Fig. 6, clearly shows Betula pendula's range overlapping with the Georgian trap locations.

We wanted to express in the first sentence of the Betula section, that other species than B. pendula and B. pubescens grow in Georgia and Turkey. Traps from both areas were included. Rephrased: Some other Betula species can appear around traps in the Caucasus and Turkey.

Page 16, line 4: There are at 6 species of Corylus in the Caucasus and C. colurna occurs in both Georgia and Turkey, so perhaps add "…and other species" here.

done

5: What do you mean by "as discussed in the main text"?

deleted

8: Replace "small" with "low"

done

Page 17, line 7: Italics for Picea abies

done

Page 19, line 10: Add "The" before "highest". Change "Balkan" to "Balkans"

done

12: "seem too high" – this is interpretation and misplaced in the Results section

deleted

Page 20, line 4: Lagodekhi misspelled

done

5: Semicolon after "Georgia"

done

6: The highest…

done

Page 21, line 4: This is the first time taphonomy is mentioned in the paper, which seems a significant oversight.

According to the major comment mentioned in the Introduction.

7: Add comma after "locations"

done

11: "On the regional scale PAR" – replace with "On a regional scale, PAR"

done

13: If latitude influences primary productivity (and thereby pollen production), then surely elevation has a similar effect?

Yes, we added sentences on the end of section 4.2.

15-16: This seems to be saying the same thing as lines 8-11. Or do you mean average PAR, or regional PAR here?

Yes

Page 22, line 2: How do the PARs for local vs long-distance presence from that study compare to the thresholds in this paper?

Sentance added: "Our general threshold 80 grains cm-2 y-1 is slightly lower than their range for Pinus and Betula in arctic-apline zone 100-200 grains cm-2 y-1."

8: Replace "larger" with "higher"

done

– and explain how the fall speed influences pollen thresholds.

Reference to Table S6 added.

It's hard to tell whether the data support the authors' claims about thresholds here as the Fagus results (Fig. 4) do not cover the same distance range as the Corylus results. Consider revising this statement.

Statement revised.

18-20: Remind the reader in a few words why fossil data from these areas were considered unreliable.

Reliable fossil PAR record is produced in a stable sedimentation environment. We found only fuzzy record and we do not have detailed knowledge of the sedimentation processes as mentioned on the end of the first paragraph of 4.4.,

21: Split "for the"

rewritten

22: Is PAR plural here?

rewritten

24: Comma after "percentages"

done

Page 23, line 2: Please provide a source for the statement that nitrogen increases pollen production in other tree species independently of changes in forest composition.

We added Pers-Kamczyc et al 2020.

Please provide details of the $CO_2$ experiment – were the levels of $CO_2$ comparable to the current climate, i.e. is the recent increase in atmospheric $CO_2$ sufficient to explain the recent increase in pine pollen in the Brandenburg forests?

They fumigated with 200 microlitres per liter which corresponds to 200 ppm more than the environment. This difference is twofold higher than the rise of $CO_2$ from 1900 (100 ppm). $CO_2$ itself can not explain that, you have to take into an account the growing volume, nitrogen enrichment from fertilization and possible interactions.

12: Make "percentage" plural

done

23: Comma after "sites"

done

Page 24, line 1: "in boreal region result above" – seems to be a word missing here

swapped with "are"

3: This section (4.4) has many sentences starting with "Nevertheless, Although, Also, Despite" – this stream of contradictions makes the argument seem disjointed.

done

6: What are some examples of these modern processes? This statement assumes that these are common knowledge.

"such as pollen from trapped insects"

8: Replace "stringer" with "stronger"

done

13: How the "best available" sites were chosen should be elaborated in the Methods

added

15: Comma after "available"

done

20: Replace "high" with "large"

done

25: In this context it might be worth referring to Tauber's experiments with roofed and unroofed traps, where the roofed traps would have presumably avoided any direct pollen fall.

I think that it is quite clear, that when leaves fall on the opening, that less or no pollen is coming.

28: Comma after "dataset"

done

29: Replace "is" with "are"

done

Page 25, line 7: Double check Conclusions once other changes have been made (also Abstract)

Sentence to abstract added.

7: One instance of "that" needs to be deleted

done
S1 and S2 – the axes are labelled as % while the data seem to be proportions
Corrected
S3 Carpinus – caption is missing (a).
We cannot find any missing caption for (a)-
For part (d), the maximum for site "Sum" is not highlighted (intermediate values highlighted instead)
Yes, Carpinus at Suminko is listed in Table S3, where you can find all cases, where the second highest class of fossil PAR was used for the link with modern analogues.
S3 Fraxinus – "demining tree" perhaps should be "demanding tree"
Changed

---

## Author Response (AR3)

Congratulations to the authors for the extensive revision of their manuscript. It now reads more succinctly and coherently.

Thank you, Simon! We have rewritten LDT sections 1.3, 3.3., 4.2 and Conclusions; added sentences about taphonomy and re-checked the English.

The revised Introduction sets up the main arguments much more effectively. There are a few remaining issues that can be easily resolved:

1. The conclusion that, since fossil PARs and trap PARs show the same range of values, "indicates that there are no major biases hampering the application of the PMP Database data as a modern reference to interpret the fossil record" [p. 24, line 15]. This conclusion may be overly simplistic, given the limitations listed in section 4.4. I think a more useful conclusion would be for the authors to indicate what further research is required in order to make the link between fossil and trap PARs, or what considerations are needed in making such links. Otherwise we could find researchers erroneously linking their fossil PARs from large lakes to pollen traps that have completely different source areas and taphonomic processes. There is no doubt the trap PARs are very useful, but they must be used with care!

First paragraph of conclusions:
Comparison of the mean annual PAR from traps and fossil sites showed similar ranges for the common European trees at the continental scale. Fossil PAR values can be linked to modern analogues in Europe, opening up possibilities for using fossil PARs to inform on past changes in plant biomass and primary productivity. However, careful selection of fossil sites is necessary to a avoid biases of absolute pollen deposition in the fossil record, which may be caused by lake internal processes such as focusing or the addition of pollen from the catchment or bank erosion.

2. There remains confusion about the reported thresholds for long-distance-transported pollen (LDT). The abstract reports 30 grains/cm2/yr, the conclusion reports < 80 grains/cm2/yr, but in the main text and figures there are different values for different taxa. Why not provide taxon-specific thresholds, which will be much more reliable? While 80 grains is probably okay for Fagus, it is clearly inappropriate for Tilia, which is insect-pollinated and a poor pollen disperser. Application of the 80-grains threshold could lead researchers to mistakenly infer absence of Tilia, when in fact it was present.

We provide species specific threshold in conclusions:

The following thresholds of PAR for were obtained for a distance of 200 km from a distribution limit: Quercus (50), Carpinus (39), Fraxinus (20), Corylus (15), Tilia (5) and Picea (0.3). The obtained threshold of 80 grains cm-2 y-1 for Fagus is likely too high.

, however, in abstract we included more general sentence:

The threshold of this long distance transport for individual species is generally below 60 grains cm^-2^ y^-1^.

3. The same applies to the reporting "Regional forest cover >80% is indicated by >3200 tree pollen grains grains cm-2 y-1" – this value is obtained from the minimum trap values, not from the average, so it is unclear how it could be applied in the fossil context

OK, we describe the relationship followingly:

"In treeless vegetation PAR values of at least 140 grains cm$^{-2}$ year$^{-1}$ are found and with each 10% of forest cover tree PAR increases by 400 grains cm$^{-2}$ year$^{-1}$ at least."

and in conclusions we add:

"Minimum PAR values rise with increasing forest cover within 10 km of the trap, while the maximum values are determined by local site conditions. Fossil PAR data may therefore be of limited use when aiming to reconstruct regional forest cover. At least 140 grains cm-2 year-1 of tree pollen may be found in treeless vegetation and with each 10% of forest cover tree PAR increases by at least by 400 grains cm-2 year-1. "

4. Figures 6-13 (plus figures in the Supplement) show "Crossed squares indicate that pollen of the taxon was not found in any trap from the area", yet many of these crossed traps actually seem to have pollen in them, as shown from coloured symbols in the 1-dimensional clustering. How is this possible?

corrected to: "Crossed squares indicate that pollen of the taxon was absent from at least in one trap from the area."

Specific comments by page:line

[Title] Change semicolon (;) to a colon (:) or hyphen (-)

Changed to hyphen

[2:7-8] "grains grains" -> grains

done

[2:9] Mention which taxa this 30-grains threshold applies to, otherwise we could find this threshold being applied incorrectly. The Results section now includes taxon-specific thresholds (mostly 80 grains, section 3.3 p. 12), which are more useful than a blanket value.

Pollen traps situated beyond 200 km of the distribution of a given tree species still collect occasional pollen grains of that species. The threshold of this long distance transport for individual species is generally below 60 grains cm$^{-2}$ y$^{-1}$.

[2:11] This sentence is hard to understand without a fuller explanation. Do you mean something like "For temperate taxa, modern analogues for fossil PARs are generally found downslope or southward of the fossil sites"?

changed

[2:13] "data is… aids" -> data are… aid

done

[2:19] "became the widely" -> has become the most widely

done

[3:23] "Europe" -> European

done

[3:28] the number 3) is missing here

done

[4:1] consider deleting "in addition to determining biomass" as this makes this question confusing

deleted

[4:9] "productivity than" -> productivity, then [note spelling]

done

[4:12] This section is very brief and would benefit from additional justification. Perhaps consult Froyd 2005 https://doi.org/10.1890/04-0546 and literature therein for a discussion of rational limits and the limitations of traditional palynological methods in detecting tree migration.

expadned to:

Identification of the local presence of taxa in the past by comparison of different proxies, produces ambiguous results. Some studies show that the rise of PAR is able to mirror the first occurrence of a macrofossil [e.g. @giesecke2005holocene], others show an increase in PAR and percentage values thousands years after the first appearance of stomata [@froyd2005; @parshall_documenting_1999]. So, the modern comparisons of PAR thresholds and recent vegetation are needed.

[4:19] consider "analogues" instead of "analogies"

done

[4:21] "environmental conditions assemblages" -> environmental conditions

done

[4:24] add a comma after "obtained"

added

[5:17] should "regional forest cover" be "regional biomass" to fit with section 1.2?

done

[6:16] Lagodekhi is misspelled here

done

[10:6] "in the Holocene perspective" -> from a Holocene perspective

done

[12:17] "a 3% wide bins of the forest" -> in 3% wide bins of forest

done

[the wording in this sentence is very unclear and it may help to rewrite it in simpler terms, perhaps 2 sentences; it's also unclear why 80% forest cover is important here – can you explain why this value is presented rather than 0% and 100%, which might be more useful?]

The traps with the lowest PAR per each 3%-wide bin of forest cover provide a regression model. The relationship predicts a PAR of at least 140 grains cm^-2^ year^-1^ in a treeless vegetation. With each 10% of forest cover within 10 km of the trap, tree PAR increases at least by 400 grains cm^-2^ year^-1^.

[12:20] "distribution limit" -> distribution limits

done

[12:26] "that one trap" -> that only one trap

done

[12:31] "in average" -> on average

done

[14:5] "> 0.05" = non-significant; do you mean < 0.05 (i.e. significant)?

yes, I mean > 0.05 = non-significant and < 0.05 is significant. However, the null hypothesis of the t-test is that the mean of trap PAR and the mean of fossil PAR are equal, since the p-value is > 0.05, we can not reject the null hypothesis (means are equal or similar).

[14:17] "only analogous trap area with PAR for fossil sites" -> only trap area analogous to fossil sites

done

[16:Fig. 6 caption] "Mean modern PAR averaged" -> Mean modern PAR for selected tree taxa averaged;

done

 "falling within the pollen taxa" -> falling within the pollen trapping area [?]

changed to "from the pollen taxa"

[16:5] replace comma with semicolon

done

[20:3-5] I suggest separating this into 2 sentences – one about using PARs to interpret fossil signals and another detailing the taphonomic differences.

done

The authors' implied claim that taphonomy is not important in this comparison needs additional justification.

Justification added: Processes involving differences in the efficiency of capture and deposition of pollen on a surface are important explaining local variability, while the added uncertainty is generally smaller than the overall signal. On the other hand lake internal processes like focusing, bank erosion

or riverine pollen input may alter the fossil signal substantially and here careful site selection and site specific interpretation are needed to allow comparisons [@giesecke2008].

[20:15] CO2 is one factor, but temperature changes have also been shown to have an influence on pollen production (e.g. see this example from Switzerland: https://doi.org/10.1007/s00484-008-0159-2). Perhaps this could be incorporated?

An increase in primary productivity and pollen production has been shown in a carbon dioxide fertilization experiment [@wayne2002] and in an increase of the temperature due to global warming [@frei2008]. Both factors support the interpretation that average PAR of the same species may vary due to environmental parameters determining its productivity.

[20:16] remove "its"

done

[21:4] "arctic-apline" -> arctic-alpine

done

[21:5] "greater distances" – do you mean greater than in Seppä & Hicks 2006? It would help the reader to follow your reasoning if you provided the distances.

substituted by: "larger spatial scale and less precise vegetation data"

[21:12] "Corylus has a lighter pollen grain than Fagus, which can travel more easily over large distances" – this sentence means that Fagus pollen travels further than Corylus, but I don't think that is the intended meaning. Perhaps "Corylus has a lighter pollen grain than Fagus that can travel more easily over large distances" or "Corylus pollen is lighter than Fagus pollen and able to travel greater distances"

changed to: "Corylus pollen is lighter than Fagus pollen and able to travel greater distances (Table S5)." Additionally we removed the comparison with the characteristic radius.

[21:13] "LDT for Picea results too low and Fagus too high" – unclear sentence. Is there a verb? This seems an important observation but lacks justification. Please elaborate.

rewritten

[21:17] "in average" -> on average [x 2]

done

[21:18] "by too leptokurtic" -> by the overly leptokurtic

done

[21:19] "that it underestimates dispersal of pollen with a large grain" -> that the model underestimates dispersal of large pollen grains

done

[21:21] "with the" -> with

done

[21:22] "the actual growth density" – do you mean "biomass"?

done

[21:22] "because of the worse climatic" -> because of unsuitable climatic

done

[23:4-5] "…individual sites. Unfortunately, the details cannot be discussed here. However…" – this structure is very jumpy. Why not simply say "…individual sites. For instance…"

done

[23:14] delete "Nevertheless" here

done

[23:17-19] avoid successive sentences starting with "Also… Nevertheless… Despite…" – this indicates poor structure.

removed

[23:25] delete "respectively" [not needed here];

done

the comment that the fossil PARs and trap PARs are have the same range of values seems lost here – what are the authors trying to say? Should we expect similar values when the taphonomic processes are different?

Taphonomic explanation added, see above.

[23:31ff] this paragraph repeats information from the Methods section and could be removed.

removed, information about the reduced PAR moved to Methods.

[24:7] "meters" -> metres [to conform with UK spelling used throughout the manuscript]

done

[24:14-16] "PAR from traps and fossil sites showed similar ranges for Abies, Alnus, Betula, Carpinus, Corylus, Fagus, Fraxinus, Picea, Pinus, Quercus and Tilia at the continental scale. This indicates that there are no major biases hampering the application of the PMP Database data as a modern reference to interpret the fossil record" – this assumes that pollen production has not changed through time, that the taphonomy of traps and lakes/wetlands is the same, that the vegetation mosaic is the same now as it was in the past, and that all the PAR measurements are accurate. Section 4.4 indicates that these assumptions are not met. What about a different conclusion, e.g. that fossil PAR values can be linked to modern analogues in Europe, opening up possibilities for using fossil PARs to reconstruct past plant biomass and primary productivity?

To sect 4.4 we added: lake internal processes often lead to PAR values exceeding modern ranges. Such biased fossil PAR estimates can in turn be used to elucidate the sedimentation history.

Conclusions substantially rewritten.

[24:18-19] "Minimum values suggest that an 80% forest cover within 10 km of the trap results in PAR above 3200 tree pollen grains cm-2 year-1." – as mentioned earlier [12:17] reporting these values seems strange, especially as a major conclusion of the paper – how can they be applied when they derive from the minimum PAR, which would not be known in the fossil context?

Conclusions:
„Fossil PAR data may therefore be of limited use when aiming to reconstruct regional forest cover. At least 140 grains cm-2 year-1 of tree pollen may be found in treeless vegetation and with each 10% of forest cover tree PAR increases by at least by 400 grains cm-2 year-1."

[24:20] the threshold here is different from the one in the Abstract

In treeless vegetation PAR values of at least 140 grains cm^-2^ year^-1^ are found and with each 10% of forest cover tree PAR increases by 400 grains cm^-2^ year^-1^ at least.

[Supplement] since page numbers are not a limitation in the supplement, why not arrange the text in the section "Taxa specific linkage of the highest average PAR at fossil sits [*sites?] with individual trap values" so that each figure is accompanied by the relevant text? For example, the Carpinus plot could appear on the same page as the Carpinus text.

Thank you for the idea, that was original intention.

---

## Author Response (AR4)

[revised manuscript text omitted]

**Correspondence:** Vojtěch Abraham (vojtech.abraham@gmail.com)

**Abstract.** The collection of modern, spatially extensive pollen data is important for the interpretation of fossil pollen assemblages and the reconstruction of past vegetation communities in space and time. Modern datasets are readily available for percentage data but lacking for pollen accumulation rates (PAR). Filling this gap has been the motivation of the pollen monitoring network, whose contributors monitored pollen deposition in modified Tauber-traps for several years or decades across Europe. Here we present this monitoring dataset consisting of 351 trap locations with a total of 2742 annual samples covering the period from 1981 to 2017. This dataset shows that total PAR are influenced by forest cover and climate parameters, which determine pollen productivity and correlate with latitude.  In treeless vegetation PAR values of at least 140 grains cm$^{-2}$ year$^{-1}$ are found and with each 10% of forest cover tree PAR increases by 400 grains cm$^{-2}$ year$^{-1}$ at least, Pollen traps situated beyond 200 km of the distribution of a given tree species still collect occasional pollen grains of that species.  The threshold of this long distance transport  generally below 60 grains cm$^{-2}$ y$^{-1}$ . Comparisons between modern and fossil PAR from the same regions show similar values.  For temperate taxa, modern analogues for fossil PARs are generally found downslope or southward of the fossil sites. While we do not find modern situations comparable to fossil PAR values of some taxa (e.g., *Corylus*). $CO_2$ fertilization and land use may cause high modern PAR that are not documented in the fossil record. The modern data  are now publicly available in the Neotoma Paleoecology Database and aids interpretations of fossil PAR data.

*Copyright statement.* The article is distributed under the Creative Commons Attribution 4.0 License. Unless otherwise stated, associated published material is distributed under the same licence.

[revised manuscript text omitted]